# Size Transferability of Graph Convolutional Networks across Sparsity: A Generalized Graphon Perspective

**Qinji Shu** [1]  **Hang Sheng** [1]  **Feng Ji** [2]  **Hui Feng** [1 3]  **Bo Hu** [1 3]

## Abstract

Size transfer scales Graph Convolutional Networks (GCNs) by applying models trained on sampled subgraphs to larger target graphs. However, existing theoretical guarantees are typically confined to dense graphs or restricted sparsity regimes, failing to cover the arbitrary sparsity of real-world networks. To bridge this gap, we introduce the Generalized Graphon Convolutional Network (GWCN) based on the generalized graphon theory. Unlike the classical graphon limit which vanishes in sparse settings, GWCN employs stretching to construct a non-trivial limit that preserves topological structure. We derive an explicit transfer error bound that decomposes into size-dependent and density-dependent components, providing a unified guarantee across arbitrary sparsity levels. Empirical results on real-world networks corroborate our findings, demonstrating that transfer error vanishes as graph size increases and edge density decreases.

## 1. Introduction

Graph convolutional networks (GCNs) (Defferrard et al., 2016; Kipf & Welling, 2017) have demonstrated robust performance across real-world network tasks (Wang et al., 2019; Zhang et al., 2019; Xia et al., 2023; Sharma et al., 2024). However, training on large graphs incurs high computational overhead. To address this, size transfer is widely adopted, where GCNs are trained on downsampled smaller graphs and transferred to larger targets (Chen et al., 2018; Chiang et al., 2019a; Yao & Li, 2021).

---

[1]College of Future Information Technology, Fudan University, Shanghai, China [2]School of Electrical and Electronic Engineering, Nanyang Technological University, Singapore [3]the State Key Laboratory of Integrated Chips and Systems, Fudan University, Shanghai, China. Correspondence to: Feng Ji <jifeng@ntu.edu.sg>, Hui Feng <hfeng@fudan.edu.cn>.

*Proceedings of the 43rd International Conference on Machine Learning*, Seoul, South Korea. PMLR 306, 2026. Copyright 2026 by the author(s).

Despite the empirical success of size transfer, establishing rigorous theoretical guarantees remains limited (Cordonnier, 2024; Morris et al., 2024; Levin et al., 2025). Most analyses of GCN transferability employ graph limit theory, where a sequence of graphs converges to a continuous limit as the graph size increases (Janson, 2010; Lovász, 2012; Backhausz & Szegedy, 2022; Ji et al., 2024). A foundational tool in this domain is the classical graphon (Ruiz et al., 2020; Magner et al., 2022; Ruiz et al., 2023; Maskey et al., 2023). While effective for uniform sampling strategies, it inherently preserves edge density at $\Theta(1)$ as the graph grows, implying quadratic edge growth. This fundamentally characterizes the dense case (Orbanz & Roy, 2015; Borgs et al., 2018).

Real-world networks typically exhibit sparsity (Barabási & Albert, 1999; Albert & Barabási, 2002), resulting in edge density variations that hinder size transfer. Sparsity implies limited node connectivity. Specifically, the number of edges grows significantly slower than the maximum possible connections, causing edge density to diminish as the graph scales (Borgs et al., 2018) (see Fig. 1). In the context of size transfer based on topological similarity, dense graphs maintain stable density, requiring models to account primarily for size differences. Conversely, in sparse graphs, varying density introduces an additional discrepancy that impedes effective transfer.

Critically, the downsampling process in size transfer complicates this sparsity profile. Depending on the sampling strategy, subgraph edge densities often diverge not only from the original graph but also from each other (Zeng et al., 2020; Chiang et al., 2019b; Leskovec & Faloutsos, 2006). This leads to varying sparsity, characterized by fluctuating levels rather than a single, predictable trend. Consequently, analyzing GCN transferability across these inconsistent sparsity regimes presents a substantial obstacle.

The inherent sparsity of real-world networks and the varying sparsity phenomenon induced by downsampling, create difficulties for GCN analysis. Some studies of GCNs on sparse graphs bypass explicit graph generation by directly sampling operators (Levie et al., 2021; Le & Jegelka, 2023). Therefore, topological properties like edge density cannot be directly analyzed. Other works guarantee convergence or transferability only under restricted sparsity regimes, such

*Table 1.* Comparison of theoretical frameworks for GCN transferability.

| Works | Sparsity Support (Edge Density) | Convergence | Range of Applicability for $k_n$ in Fig. 1 |
|---|---|---|---|
| Classical Graphon (Ruiz et al., 2020; 2023) | Dense ($\Theta(1)$) | Yes | $k_n = \Theta(n)$ |
| Unbounded Graphon (Maskey et al., 2023) | Dense ($\Theta(1)$) | Qualitative | $k_n = \Theta(n)$ |
| Continuous Space (Levie et al., 2021) | Relatively Sparse ($\Omega(\frac{\log n}{n})$) | Yes | Inexplicit |
| Bounded Graphon (Keriven et al., 2020; 2021) | Relatively Sparse ($\Omega(\frac{\log n}{n})$) | Yes | $k_n = \Omega(\sqrt{n \log n})$ |
| Geometric Graph (Wang et al., 2024) | Relatively Sparse ($\Omega(\frac{\log n}{n})$) | Yes | $k_n = \Omega(\sqrt{n \log n})$ |
| Operator and Graphop (Le & Jegelka, 2023) | Implicitly Arbitrary | Yes | Inexplicit |
| Dense-Sparse Model (Ruiz et al., 2024) | Arbitrary | **No** | $\forall k_n$ |
| **Ours (Generalized Graphon)** | **Arbitrary** | **Yes** | $\forall k_n$ |

*Note: Sparsity Support* denotes sparsity constraints. *Range of Applicability for $k_n$ in Fig. 1*: Based on the example in Fig. 1 where sparsity is controlled by the clique size $k_n$, this column illustrates that **weaker sparsity constraints cover a broader regime of graph topology**, denoted by a wider range of $k_n$. Terms like "Implicitly Arbitrary" and "Inexplicit" refer to **directly sampling operators**, where graph structures (like edge sets) are not explicitly generated. *Convergence* indicates whether the error vanishes; "Qualitative" denotes convergence without an explicit decay rate.

as $\Omega(\log n/n)$ for graph size $n$ (Keriven et al., 2020; 2021; Wang et al., 2023; 2024). Consequently, when considering transfer between graphs across different sparsity levels, including higher sparsity, existing results do not address this case. To broaden the transferability analysis of sparse GCN, it is necessary to establish a theoretical guarantee that holds across arbitrary sparsity levels.

In this paper, we tackle the problem of sparse GCN transfer across varying sparsity, inspired by the generalized graphon model (Borgs et al., 2018; 2019; Ji et al., 2024). Our approach guarantees transferability across arbitrary sparsity and provides explicit transfer bounds. The generalized graphon model generates sparse graphs via double-sampling and employs stretching to ensure a double-convergent non-zero limit. Stretching amplifies the structures of sparse graphs to vanishing effects caused by sparsity (Borgs et al., 2018; 2019; Ji et al., 2024). Building upon these, we introduce the Generalized Graphon Convolutional Networks (GWCNs) as the meaningful limit of GCNs for arbitrary sparsity. Because these graph sequences share a common limit regardless of sparsity variations, we are able to derive a transfer error bound across arbitrary sparsity levels.

Our contributions to this work are as follows:

- We leverage the generalized graphon framework to model the varying sparsity induced by downsampling in size transfer. This enables us to derive a meaningful limit for GCNs, which remains stable across sequences with fluctuating densities.

- We introduce the generalized graphon convolutional network (GWCN) as the non-zero limit of GCNs for arbitrary sparsity, providing a meaningful reference for convergence and transferability analysis.

- We prove that GCN transferability holds across arbi-

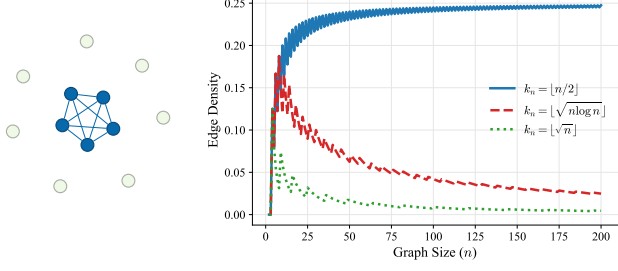

*(a) $G_n : k_n = 5$.*   *(b)* Edge Density Curves.

*Figure 1.* **Edge density of the example graph sequence $\{G_n \mid k_n\}$.** In each graph $G_n$, a subset of $k_n$ nodes forms a fully connected subgraph, while the remaining nodes are isolated. By setting $k_n$ to $\Theta(n)$ or $o(n)$, the sequence corresponds to the dense case (e.g. $k_n = \lfloor n/2 \rfloor$) or the sparse case (e.g. $k_n = \lfloor \sqrt{n \log n} \rfloor$, or $k_n = \lfloor \sqrt{n} \rfloor$).

trary sparsity, and establish an explicit upper bound on the transferability error. This bound reveals that the transfer error is governed by topological information, specifically graph size and edge density.

## 2. Related Work

The widespread applicability of GCNs originates from their ability to propagate and aggregate information on irregular graph structures (Defferrard et al., 2016; Kipf & Welling, 2017; Zhang et al., 2019). However, a theoretical question arises: how to quantify the difference between different graphs? This is generally achieved by modeling random graphs and establishing convergence within models (Cordonnier, 2024; Morris et al., 2024; Levin et al., 2025), such as the classical graphon model for dense case (Lovász, 2012; Ruiz et al., 2021a), and the generalized graphon model for sparse case (Borgs et al., 2018; 2019; Ji et al., 2024).

**Transfer in Dense Graphs**. Within dense graph sequences employed for analyzing GCN transferability, the classical graphon is a widely adopted framework (Ruiz et al., 2020; Magner et al., 2022; Cai & Wang, 2022; Ruiz et al., 2023; Maskey et al., 2023; Cerviño et al., 2023; Velasco et al., 2024). Measured by cut norm and cut distance, the classical graphon ($W : [0,1]^2 \rightarrow [0,1]$) provides a functional representation of graph topology. It serves as a generative model for dense graph sequences and the limit of such sequences (Janson, 2010; Lovász, 2012). For a given graphon $W$, the expected edge density remains constant $\|W\|_1$, making edges scale at a dense rate $n^2\|W\|_1$. Ruiz et al. (2021a) extend the graphon model by incorporating graphon signals ($X : [0,1] \rightarrow \mathbb{R}$), allowing for a joint sampling of graph topology and node features, and further establish graphon signal processing framework. In their subsequent work (Ruiz et al., 2020; 2023), the convergence of graph sequences is extended to GCN sequences, then the GCN transferability is analyzed. Furthermore, Maskey et al. (2023) broaden the classical graphon to value-unbounded graphon to accommodate operators like Laplacian, though still within the dense regime. Recently, Shu et al. (2025) established finite-size convergence bounds, overcoming the constraint of requiring sufficiently large graphs in prior works. Based on the classical graphon, the graph classification capability (Magner et al., 2022) and the transference training efficiency (Cerviño et al., 2023) of GCNs are analyzed, the conclusions on GCNs are extended to other graph neural networks (Cai & Wang, 2022; Velasco et al., 2024).

**Transfer in Sparse Graphs**. Since real-world networks are typically sparse with decreasing edge densities, sparse graph models are adopted in analyzing the transferability of GCNs (Keriven et al., 2020; 2021; Wang et al., 2023; 2024; Ruiz et al., 2024). In Keriven et al. (2020), sparse graph sequences are constructed by multiplying the classical graphon with a decreasing sparsity factor $\alpha_n$. As the graph size grows, the graphon tends to zero, resulting in vanishing edge densities. Instead of the zero limit, they introduce a normalized Laplacian operator divided by the sparse degrees, yielding a relatively sparse convergence conclusion. Keriven et al. (2021) normalize adjacency matrix by increasing edge weights reversely proportional to sparsity ($\alpha_n^{-1}\mathrm{Ber}(\alpha_n W)$), ensuring that expected edge weights remain stable. Although sparsity weakens connectivity, their enhanced weights reinforce the propagation along existing edges. Similarly, Wang et al. (2023; 2024) construct relatively sparse geometric graphs with increasing weights inversely proportional to sparsity, and study the GCN convergence to manifold neural networks.

In general, these works analyze convergence or transferability under models with restricted sparsity levels, such as relative sparsity or lower. Other modeling approaches obtain graph operators directly through discretization on geometric

spaces or continuous operators, without generating explicit random graphs (Levie et al., 2021; Le & Jegelka, 2023). Since graph structures are not explicitly realized in these frameworks, topological properties like edge density cannot be directly analyzed. Furthermore, while Ruiz et al. (2024) utilize a kernel function to construct sparse graphs without strict sparsity restrictions, their GCN spectral convergence error does not vanish but approaches a constant.

To clarify the landscape of existing theories, we provide a systematic comparison in Table 1. As summarized in the table, prior frameworks are either limited to restricted sparsity regimes (e.g., dense or relative sparsity), lack explicit topological analysis due to non-generative modeling, or fail to guarantee vanishing transfer errors. This highlights the need for a theoretical guarantee that holds across arbitrary sparsity levels while ensuring error convergence.

## 3. The Sparse Random Graph Model

In this paper, we consider a sparse random graph model based on the generalized graphon framework (Borgs et al., 2018; 2019; Ji et al., 2024). This line of work is closely related to the graphex framework (Borgs et al., 2021), which provides a comprehensive limit object capturing structural properties of sparse graphs. Given that the convolutional operator is the key to GCNs, we adopt the signal processing framework from Ji et al. (2024), which is precisely tailored to analyze the limit of sparse graph sequences.

In this model (Ji et al., 2024), sparse graphs are generated through a double-sampling procedure: first, a sparse classical graphon sequence $\{W_m\}$ is sampled, and then sparse graphs $\{G_{m,n}\}$ are sampled from this sequence. This construction enables the generation of graph subsequences with arbitrary sparsity (see Fig. 4),

$$\text{Sparsity(Graph size)} = \text{Expected edge density.} \quad (1)$$

Here, edge density is a topological property defined as the ratio of the realized number of edges $e(G_n)$ to the total possible edges $n^2/2$, and sparsity refers to the decay function of the expected edge density as graph size $n$ increases.

**Definition 3.1** (Generalized graphon and signal (Borgs et al., 2018; 2019; Ji et al., 2024))**.** A generalized graphon $W_{\mathbb{R}_+}(u,v)$ is a symmetric function: $\mathbb{R}_+^2 \rightarrow [0,1]$. A generalized graphon signal $X_{\mathbb{R}_+}(u)$ is a function: $\mathbb{R}_+ \rightarrow \mathbb{R}$.

We require both $W_{\mathbb{R}_+}$ and $X_{\mathbb{R}_+}$ to be $L^1$-integrable and bounded, ensuring that stretching and convolution operations are well defined. The topological variation between generalized graphons is measured using the extended cut norm and cut distance (Borgs et al., 2018; 2019). By extend-

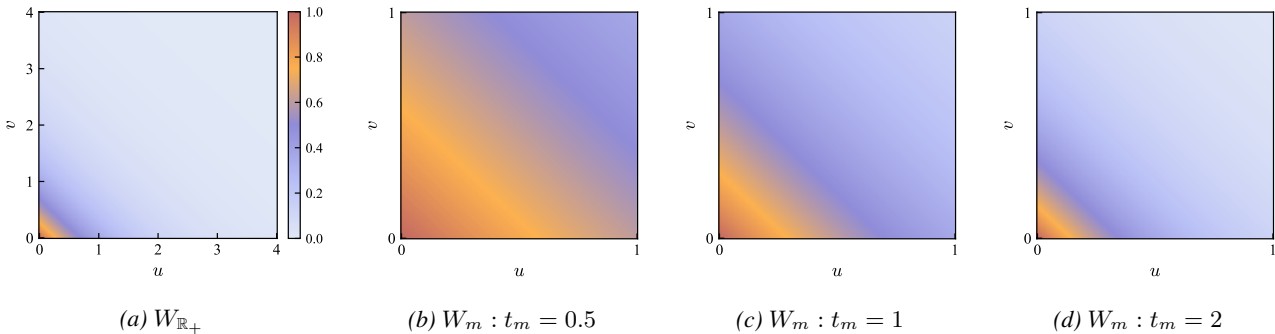

*(a) $W_{\mathbb{R}_+}$*      *(b) $W_m : t_m = 0.5$*      *(c) $W_m : t_m = 1$*      *(d) $W_m : t_m = 2$*

*Figure 2.* **Example of sparse graphon sequence:** (a) $W_{\mathbb{R}_+}$ is the original generalized graphon, (b)-(d) illustrate $W_m$, obtained by truncating the domain $[0, t_m]^2$ of $W_{\mathbb{R}_+}$ and rescaling it to $[0, 1]^2$.

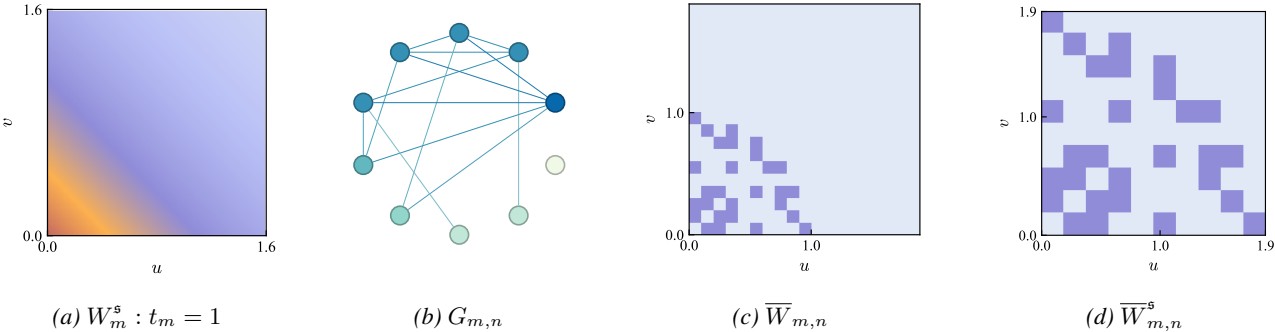

*(a) $W_m^{\mathfrak{s}} : t_m = 1$*      *(b) $G_{m,n}$*      *(c) $\overline{W}_{m,n}$*      *(d) $\overline{W}_{m,n}^{\mathfrak{s}}$*

*Figure 3.* **Example of stretching:** $G_{m,n}$ is a random graph sampled from $W_m$, its adjacency matrix is transformed to corresponding classical form (c) and stretched form (d), the latter converges to stretched graphon (a).

ing the integration domain to $\mathbb{R}_+^2$, they are defined:

$$
\begin{aligned}
&\delta_{\square}(W_{\mathbb{R}_+,1}, W_{\mathbb{R}_+,2}) \\
&= \inf_{\phi_1,\phi_2} \sup_{U,V \subseteq \mathbb{R}_+} \left| \int_{U \times V} (W_{\mathbb{R}_+,1}^{\phi_1} - W_{\mathbb{R}_+,2}^{\phi_2}) du dv \right|,
\end{aligned} \tag{2}
$$

where $W_{\mathbb{R}_+,1}, W_{\mathbb{R}_+,2}$ are two generalized graphons; the sup is taken over measurable subsets $U, V \subseteq \mathbb{R}_+$, and the inf is taken over measure preserving bijection $\phi_1, \phi_2$. When applied to graphons induced by finite graphs, these bijections correspond to disregarding node ordering.

### 3.1. The Sparse Graphon Sequence

The sparse graphon sequence is constructed by truncation and rescaling. At stage $m \in \{1, 2, 3, \dots\}$, $W'_m$ is obtained by truncating $W_{\mathbb{R}_+}$ to the expanding square region $[0, t_m]^2$, setting remaining area to zero (Ji et al., 2024). The truncated generalized graphon $W'_m$ is stage-expanding. Then, $W'_m$ is rescaled to the standard domain $[0, 1]^2$ to obtain the sparse classical graphon $W_m$:

**Truncate:** $W'_m(u, v) = \begin{cases} W_{\mathbb{R}_+}(u, v), & u, v \in [0, t_m], \\ 0, & \text{others.} \end{cases}$

**Rescale:** $W_m(u, v) = W'_m(t_m u, t_m v), \quad u, v \in [0, 1].$

Similarly, the graphon signal $X_m$ is obtained by truncating $X_{\mathbb{R}_+}$ to the interval $[0, t_m]$ and subsequently rescaling it to $[0, 1]$: $X_m(u) = X'_m(t_m u) = X_{\mathbb{R}_+}(t_m u)$. Since $W_{\mathbb{R}_+}$ is required to be integrable over $\mathbb{R}_+^2$, the $L^1$-norm of the classical graphon sequence $\{W_m\}$ tends to zero at the rate of $\Theta(1/t_m^2)$, indicating sparsity, as illustrated in Figure 2.

### 3.2. The Sparse Graph Sequence

Based on the sparse graphon and signal $(W_m, X_m)$, a random graph $G_{m,n}$ is sampled following the classical graphon sampling procedure (Lovász, 2012; Ruiz et al., 2021a; Ji et al., 2024). Latent features for $n$ nodes, $\{u_1, u_2, \dots, u_n\}$, are sampled i.i.d. from uniform distributions on $[0, 1]$. Node features are then obtained as $\boldsymbol{x}_{m,n}(i) = X_m(u_i)$, and edge probabilities are given by $\mathbf{P}_{m,n}(i, j) = p_{ij} = W_m(u_i, u_j)$. Finally, edges are sampled via Bernoulli distributions:

$$
\mathbf{S}_{m,n}(i, j) \sim \text{Ber}(\mathbf{P}_{m,n}(i, j)), \quad 1 \le i \le j \le n. \tag{3}
$$

For the overall sequence $\{\{G_{m_1,n}\}, \{G_{m_2,n}\}, \dots\}$, each stage $m_l$ can generate graphs of varying sizes $\{G_{m_l,n}\}$ ($l \in \{1, 2, \dots\}$). By selecting specific graphs $\{G_{m_1,n_1}, G_{m_2,n_2}, \dots\}$ from different stages, one can construct graph subsequences with arbitrary sparsity, as shown in Fig. 4. For example, to realize a sparsity of $C/n$ with respect to graph size $n_l$, one may choose the corresponding

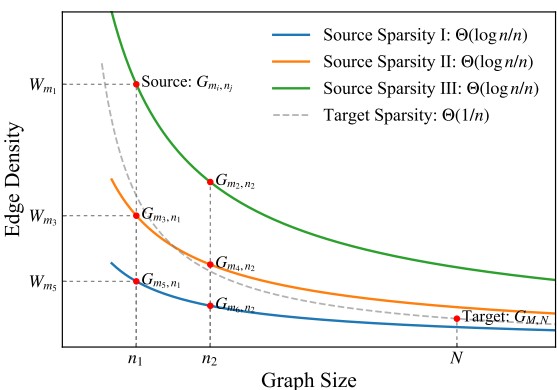

*(a)* $\Theta(\frac{\log n}{n})$ family of sparsity functions.

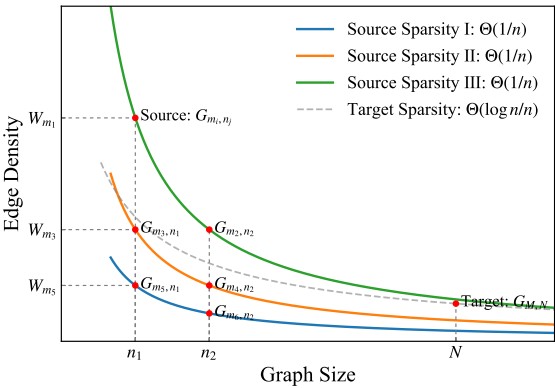

*(b)* $\Theta(\frac{1}{n})$ family of sparsity functions.

*Figure 4.* **Graph subsequences with multiple sparsity**, such as $\Theta(\frac{\log n}{n})$ in (a) and $\Theta(\frac{1}{n})$ in (b). Transfer errors are measured between the source graphs and target graph, while the target graph $G_{M,N}$ may lie beyond the sparsity line of source graphs $\{G_{m,n}\}$.

$W_m$ such that $\|W_{m_l}\|_1 = C/n_l$.

In real-world downsampling scenarios, sampling sparse subgraphs of size $n$ can produce multiple edge densities, leading to diverse sparsity levels. Previous works (Keriven et al., 2020; 2021; Wang et al., 2023; 2024) focus on models with restricted sparsity, corresponding to a single sparsity curve in Fig. 4, and guarantee transferability only for sparsity of $\Theta(\log n/n)$ or lower. However, the target graph may fall outside the sparsity regime of the sampled subgraphs, necessitating transfer analysis across different sparsity levels, as illustrated in Fig. 4.

To study the convergence of the graph sequence $\{G_{m,n}\}$, the adjacency matrix and node features are transformed into functional forms (Ruiz et al., 2020; 2023; Maskey et al., 2023). The classical induced graphon and signal are defined:

$$\overline{W}_{m,n}(u \in \mathbb{I}_i, v \in \mathbb{I}_j) = \mathbf{S}_{m,n}(i,j),$$
$$\overline{X}_{m,n}(u \in \mathbb{I}_i) = \boldsymbol{x}_{m,n}(i), \tag{4}$$

where $\{\mathbb{I}_i\}_{i=1}^n$ are $n$ equal partitions of $[0,1]$. This inducing procedure enables comparisons across graphs of different sizes and between graphs and the underlying graphon.

### 3.3. Convergence Under the Stretched form

For sparse graph sequences, the classical induced graphon $\{\{\overline{W}_{m_1,n}\}, \{\overline{W}_{m_2,n}\}, \dots\}$ converges to a trivial zero limit, rendering direct comparison with the limit meaningless. This issue is resolved via stretching (Borgs et al., 2018; 2019; Ji et al., 2024), which scales the graphon in proportion to its sparsity.

**Definition 3.2** (Stretched graphon and signal)**.** The stretched generalized graphon and signal are defined as $W_{\mathbb{R}_+}^{\mathfrak{s}}(u,v) = W_{\mathbb{R}_+}(s_R u, s_R v), X_{\mathbb{R}_+}^{\mathfrak{s}}(u) = X_{\mathbb{R}_+}(s_R u)$, where $s_R = \|W_{\mathbb{R}_+}\|_1^{1/2}$ is the stretching coefficient.

Stretching fixes the $L^1$-norm of the graphon to one, thereby amplifying structural patterns according to sparsity and counteracting vanishing effects. For a classical graphon $W_m$ and induced graphon $\overline{W}_{m,n}$, stretching is applied with zero-padding to accommodate domain differences (Borgs et al., 2018; 2019; Ji et al., 2024):

$$\overline{W}_{m,n}^{\mathfrak{s}} = \begin{cases} \overline{W}_{m,n}(s_{m,n}u, s_{m,n}v), & (u,v) \in [0, s_{m,n}^{-1}]^2 \\ 0, & \text{otherwise} \end{cases}$$
$$\overline{X}_{m,n}^{\mathfrak{s}} = \begin{cases} \overline{X}_{m,n}(s_{m,n}u), & u \in [0, s_{m,n}^{-1}] \\ 0, & \text{otherwise} \end{cases}$$
$$\tag{5}$$

where $s_{m,n} = \|\overline{W}_{m,n}\|_1^{1/2}$ is stretching coefficient that reflects graph's edge density. This operation amplifies structural characteristics in proportion to sparsity, compensating for the vanishing effect observed in sparse cases (Figure 3).

For a graph sequence $\{G_{m,n}\}$ generated by $W_m$, the stretched induced graphon $\overline{W}_{m,n}^{\mathfrak{s}}$ converges to $W_m^{\mathfrak{s}}$ under the generalized cut distance $\delta_\square$; For the sparse graphon $W_m$ truncated from $W_{\mathbb{R}_+}$, its stretched form converges to the non-zero limit $W_{\mathbb{R}_+}^{\mathfrak{s}}$ (Ji et al., 2024):

$$\lim_{m \to \infty} \lim_{n \to \infty} \overline{W}_{m,n}^{\mathfrak{s}} = \lim_{m \to \infty} W_m^{\mathfrak{s}} = W_{\mathbb{R}_+}^{\mathfrak{s}}. \tag{6}$$

Based on the double-sampling procedure, multiple graph subsequences with arbitrary sparsity can be constructed (see in Sec. 3.2). Although subgraphs $\{G_{m,n_l}|m\}$ with the same size $n_l$ but different expected edge densities are allowed under arbitrary sparsity, each $G_{m,n_l}$ is expected to approach its corresponding $W_m$. Convergence of these subsequences is guaranteed by the double convergence: at each stage $m$, graphs in a subsequence converge to the corresponding $W_m$, and the convergence of $W_m$ itself ensures overall convergence of the subsequences.

# 4. Generalized Graphon Convolutional Networks

In this section, we introduce GWCNs, which serve both as the functional forms of GCNs and as their convergent limit. Similar to the graph convolutional operator (Segarra et al., 2017; Du et al., 2018; Gama et al., 2019) in Appendix E.1, a graphon function can be regarded as an infinitely dense matrix with infinitely many nodes. The generalized graphon convolutional operator is defined as (Ji et al., 2024),

$$h(W^{\mathfrak{s}})X^{\mathfrak{s}} = \sum_{r=0}^{R} h_r T_{W^{\mathfrak{s}}}^{(r)} X^{\mathfrak{s}},$$

$$\text{for } r \geq 1, \ T_{W^{\mathfrak{s}}}^{(r)} X^{\mathfrak{s}} = \int_{\mathbb{R}_+} W^{\mathfrak{s}} \left( T_{W^{\mathfrak{s}}}^{(r-1)} X^{\mathfrak{s}} \right) du. \tag{7}$$

The generalized graphon adopts the same convolutional form, since it is defined on $\mathbb{R}_+^2$ and subsumes the stretched graphon as a special case. Because convergence of sparse graph sequences is analyzed under the stretched setting, we focus primarily on the stretched form.

Let $\Phi(W^{\mathfrak{s}}, X^{\mathfrak{s}}, \mathcal{H})$ denote the generalized graphon convolutional network (GWCN), which follows the same framework and learnable parameters $\mathcal{H}$ as the GCNs (Ruiz et al., 2020; 2021b; 2023) in Appendix E.1, with one-dimensional input features for simplicity. At the $l$-th layer, the input and output feature dimensions are $F^{l-1}$ and $F^l$, with parameters $\mathcal{H}^l \in \mathbb{R}^{F^{l-1} \times F^l}$. The channel-wise convolutional processes of GWCNs parallel that of GCNs:

$$X_{f_l}^{\mathfrak{s}} = \sigma \left( \sum_{f_{l-1}=1}^{F_{l-1}} h_{f_{l-1}, f_l}(W^{\mathfrak{s}}) X_{f_{l-1}}^{\mathfrak{s}} \right), \tag{8}$$

where $\sigma(\cdot)$ denotes the activation function, and $h_{f_{l-1}, f_l}(\cdot) = \mathcal{H}^l(f_{l-1}, f_l)h(\cdot)$ contains the learnable parameters mapping the $f_{l-1}$-th input feature to the $f_l$-th output feature. These parameters are fixed when analyzing convergence and transferability. Notably, in GWCNs each layer's output $X_{f_l}^{\mathfrak{s}}$ is obtained recursively, rather than being defined explicitly in Definition 3.2, and thus it does not directly correspond to a particular $X_{f_l}$.

# 5. Main Result

In the following sections, we first present our main result of transferability. Then we analyze in detail the convergence of the sparse graphon sequence and the random graph sequence at each stage, which together underpin our main result.

The following assumptions are considered:

**AS1.** The generalized graphon $W_{\mathbb{R}_+}$ belongs to the Schwartz space. That is, for any $k > 0$, there exits a constant $C_k$ such that $W_{\mathbb{R}_+}(u, v) \leq C_k(u+1)^{-k}(v+1)^{-k}$. Thus, it is $A_W$-Lipschitz, i.e. $|W_{\mathbb{R}_+}(u_1, v_1) - W_{\mathbb{R}_+}(u_2, v_2)| \leq$ $A_W(|u_1 - u_2| + |v_1 - v_2|)$.

**AS2.** The generalized signal $X_{\mathbb{R}_+} \in L^2(\mathbb{R}_+)$ is $A_X$-Lipschitz, i.e. $|X_{\mathbb{R}_+}(u_1) - X_{\mathbb{R}_+}(u_2)| \leq A_X|u_1 - u_2|$.

**AS3.** The activation function $\sigma(\cdot)$ is normalized-Lipschitz, i.e. $|\sigma(x_1) - \sigma(x_2)| \leq |x_1 - x_2|$, and $\sigma(0) = 0$.

In AS1, we characterize the sparsity of $W_{\mathbb{R}_+}$ by requiring it to be rapidly decreasing. For most node pairs, the connection probability is small and generally decreases as the latent feature distance increases. In AS2, the generalized signal is required to be Lipschitz continuous to guarantee local smoothness, while the $L^2$-integrability condition ensures it possesses finite energy over the unbounded domain. In AS3, the assumption on activation functions holds for most commonly used functions, such as ReLU and Tanh.

## 5.1. Transferability of GWCNs

In this section, we establish the transferability theorem of GWCNs. This result leverages the output discrepancy for distinct stretched graphons (detailed in Appendix E.3) and serves as the foundation for our transferability analysis. To fully characterize the transfer error, we then proceed to analyze the operator and signal differences mandated by the double-sampling procedure (Lemmas 5.2 and 5.3).

Consider source graph $G_{m,n}$ and target graph $G_{M,N}$ drawn from two graph subsequences exhibiting distinct sparsity, denoted as $\{G_{m_1,n_1}, G_{m_2,n_2}, \dots\}$ and $\{G_{M_1,N_1}, G_{M_2,N_2}, \dots\}$ respectively. These subsequences are constructed following the sparsity-based selection process described in Sec. 3.2.

Topologically, the sparsity is governed by the interplay between graph size and edge densities, which are effectively determined by the truncation parameters $t_m$ and $t_M$. Specifically, for graphs $G_{m,n}$ and $G_{M,N}$, the expected edge densities (corresponding to the sparsity functions in Eq.(1)) are approximated by $\|W_m\|_1 \approx \|W_{\mathbb{R}_+}\|_1/t_m^2$ and $\|W_M\|_1 \approx \|W_{\mathbb{R}_+}\|_1/t_M^2$, as illustrated in Fig. 4.

**Theorem 5.1** (Transferability of GWCNs). *Let $\Phi(W^{\mathfrak{s}}, X^{\mathfrak{s}}, \mathcal{H})$ be an L-layer GWCN with learned parameters $\mathcal{H}$, assuming $F_0 = F_1 = 1$ for simplicity. Suppose Assumptions from AS1 to AS3 hold, and the truncation lengths $t_m, t_M$ and graph sizes $n, N$ satisfy the conditions in Lemma 5.2, Lemma 5.3, and Lemma D.1. Then, for any $k' > 0$, the following bound holds:*

$$\left\| \Phi(\overline{W}_{m,n}^{\mathfrak{s}}, \overline{X}_{m,n}^{\mathfrak{s}}, \mathcal{H}) - \Phi(\overline{W}_{M,N}^{\mathfrak{s}}, \overline{X}_{M,N}^{\mathfrak{s}}, \mathcal{H}) \right\|_2$$

$$\leq C(\mathcal{H}) \left( C_{\mathbb{R}_+} \left( \frac{1}{t_m^{k'}} + \frac{1}{t_M^{k'}} \right) + \frac{L_X^2(t_m) + L_X^2(t_M)}{\|W_{\mathbb{R}_+}\|_1^{1/4}} \right)$$

$$+ C(\mathcal{H}) \left( \frac{C_{m,1}}{n^{1/2}} + \frac{C_{m,2}}{n^{1/4}} \right) + C(\mathcal{H}) \left( \frac{C_{M,1}}{N^{1/2}} + \frac{C_{M,2}}{N^{1/4}} \right), \tag{9}$$

*with probability at least* $(1 - \epsilon_{u,1})(1 - \epsilon_{b,1})(1 - \epsilon_{u,2})(1 - \epsilon_{b,2})$. *Here,* $C(\mathcal{H}) \sim L(\prod_{l=0}^{L-1} F_l)O(R^{2L})$ *depends on the network framework represented by* $\mathcal{H}$, $C_{\mathbb{R}_+}$ *depends on* $W_{\mathbb{R}_+}$ *and* $X_{\mathbb{R}_+}$, $L_X^2(\cdot)$ *is the tail integral of* $X_{\mathbb{R}_+}$, $C_{m,1}, C_{m,2}, C_{M,1}, C_{M,2}$ *are constants about* $W_m$, $X_m$ *and* $W_M$, $X_M$ *(proof in Appendix E.4).*

During size transfer, varying sparsity manifests as changes in graph size and fluctuations in edge density. Consequently, transferability across arbitrary sparsity depends on both graph sizes $(n, N)$ and their respective density scaling parameters $(t_m, t_M)$. For fixed graph sizes, increasing $t_m$ and $t_M$ (i.e., reducing expected edge densities) enhances transferability. Conversely, for a fixed expected edge density, increasing graph sizes $n$ and $N$ further improves performance. Overall, the expected edge density serves as the dominant factor, while graph size provides a secondary improvement, consistent with the double-convergence framework.

From an architectural perspective, the topological transfer error is amplified by the network's multi-layer stacking, represented by $C(\mathcal{H})$. Since our framework does not assume normalized filter responses (unlike Ruiz et al. (2020; 2023)), this architectural amplification factor grows with deeper networks, wider channels, and higher filter orders. Despite this amplification, the final transfer error still reliably converges to zero as graph sizes increase and edge densities decrease.

Here, we guarantee transferability across graph subsequences with arbitrary sparsity, while prior works (Keriven et al., 2020; 2021; Wang et al., 2023; 2024) analyze GCN only under restricted sparsity settings. In our framework, transfer errors vanish as both $t_m, t_M$ and $n, N$ increase to infinity, while the convergence errors in Ruiz et al. (2024) converge to constants. Building on this bound, we further analyze how edge density and graph size affect transferability, a perspective that Levie et al. (2021); Le & Jegelka (2023) cannot provide since they do not generate explicit graphs and thus cannot capture topological properties such as edge density.

The above GCN transfer error is basically derived from the convergence error in Theorem E.3, measured against the GWCN limit. As established in Lemma E.2, the discrepancy between GWCNs is bounded by differences in operators and signals. Therefore, in the following sections, we analyze these differences according to the double-sampling procedure in Lemma 5.2 and Lemma 5.3, which corresponds to double-convergence framework.

## 5.2. Convergence of Sparse Graphon Sequence

The convergence of a sparse graphon sequence is straightforward. Since $W_m$ and $X_m$ are obtained via truncation and rescaling, their stretched forms coincide with $W_m^{'\mathfrak{s}}$ and $X_m^{'\mathfrak{s}}$. As the truncation width $t_m$ increases, $W_m$ and $X_m$

converge to the original generalized graphon and signal.

**Lemma 5.2** (Convergence of Sparse Graphon Sequence). *Let* $(W_{\mathbb{R}_+}, X_{\mathbb{R}_+})$ *satisfy AS1 and AS2, and let* $(W_m, X_m)$ *be the sparse graphon and signal generated from them. Assume* $t_m$ *is sufficiently large such that* $\|W_m'\|_1 \geq \|W_{\mathbb{R}_+}\|_1/2$. *Then, for any* $k' > 0$,

$$
\begin{aligned}
\left\| W_{\mathbb{R}_+}^{\mathfrak{s}} - W_m^{\mathfrak{s}} \right\|_{2,2} &\leq \frac{C_{k',W}}{t_m^{k'}}, \\
\left\| X_{\mathbb{R}_+}^{\mathfrak{s}} - X_m^{\mathfrak{s}} \right\|_2 &\leq \frac{C_{k',X}}{t_m^{k'}} + \frac{L_X^2(t_m)}{\|W_{\mathbb{R}_+}\|_1^{1/4}},
\end{aligned}
\tag{10}
$$

*where* $L_X^2(t_m) = \|X_{\mathbb{R}_+}\|_{L^2([t_m, +\infty))}$ *is the tail integral of* $X_{\mathbb{R}_+}$ *on* $[t_m, +\infty)$, *constants* $C_{k',W}$, $C_{k',X}$ *depend on* $k'$, $W_{\mathbb{R}_+}$, *and* $X_{\mathbb{R}_+}$ *(proof in Appendix C).*

This result not only establishes the convergence of sparse graphons $W_m$ but also provides explicit error bounds. Under the rapid decay condition on $W_{\mathbb{R}_+}$, $W_m$ converges at an accelerated rate. In contrast, the convergence of $X_m$ is slower, governed by the tail integral over the expanding domain, and influenced by $W_m$ through the stretching.

## 5.3. Convergence of Random Graph Sequence

At each stage $m$, the random graph sequence $\{G_{m,n}\}$ has its associated limit $W_m$. This convergence holds in both the classical form and the stretched form. Moreover, convergence in the stretched form is implied by convergence in the classical form. Their notation is summarized in Table 2 in Appendix D.

**Lemma 5.3** (Convergence to Stretched Graphon). *Consider the stretched graphon and signal* $(W_m^{\mathfrak{s}}, X_m^{\mathfrak{s}})$, *and the stretched induced graphon and signal* $(\overline{W}_{m,n}^{\mathfrak{s}}, \overline{X}_{m,n}^{\mathfrak{s}})$ *generated from them. Suppose* $n$ *is sufficiently large such that* $\Delta T_{W_n} \leq \|W_m\|_1$, *then it holds*

$$
\begin{aligned}
\Delta T_{W_n^{\mathfrak{s}}} &\leq C_{\mathfrak{s}} \Delta T_{W_n} + C_{W,\mathfrak{s}} \Delta T_{W_n}^{1/2}, \\
\Delta X_n^{\mathfrak{s}} &\leq C_{\mathfrak{s}} \Delta X_n + C_{X,\mathfrak{s}} \Delta T_{W_n}^{1/2},
\end{aligned}
\tag{11}
$$

*where* $C_{\mathfrak{s}}$ *depends on the stretching coefficient* $s_m = \|W_m\|_1$, $C_{W,\mathfrak{s}}$ *and* $C_{X,\mathfrak{s}}$ *both converge to constants determined by* $W_m$ *and* $X_m$ *as* $n$ *increases (proof in Appendix D).*

Combined with Lemma D.1, which establishes an $O(n^{-1/2})$ convergence rate under the classical form, this result provides the full error bound under stretched forms. These errors arise from sampling the graph topology and node features, while stretching introduces scaling of functions and domain mismatches, captured by constants $C_{\mathfrak{s}}$, $C_{W,\mathfrak{s}}$, $C_{X,\mathfrak{s}}$. According to the double convergence in Eq.(6), these constants remain fixed while taking the limit over $n$, which is done before convergence in $t_m$, guaranteeing the overall convergence.

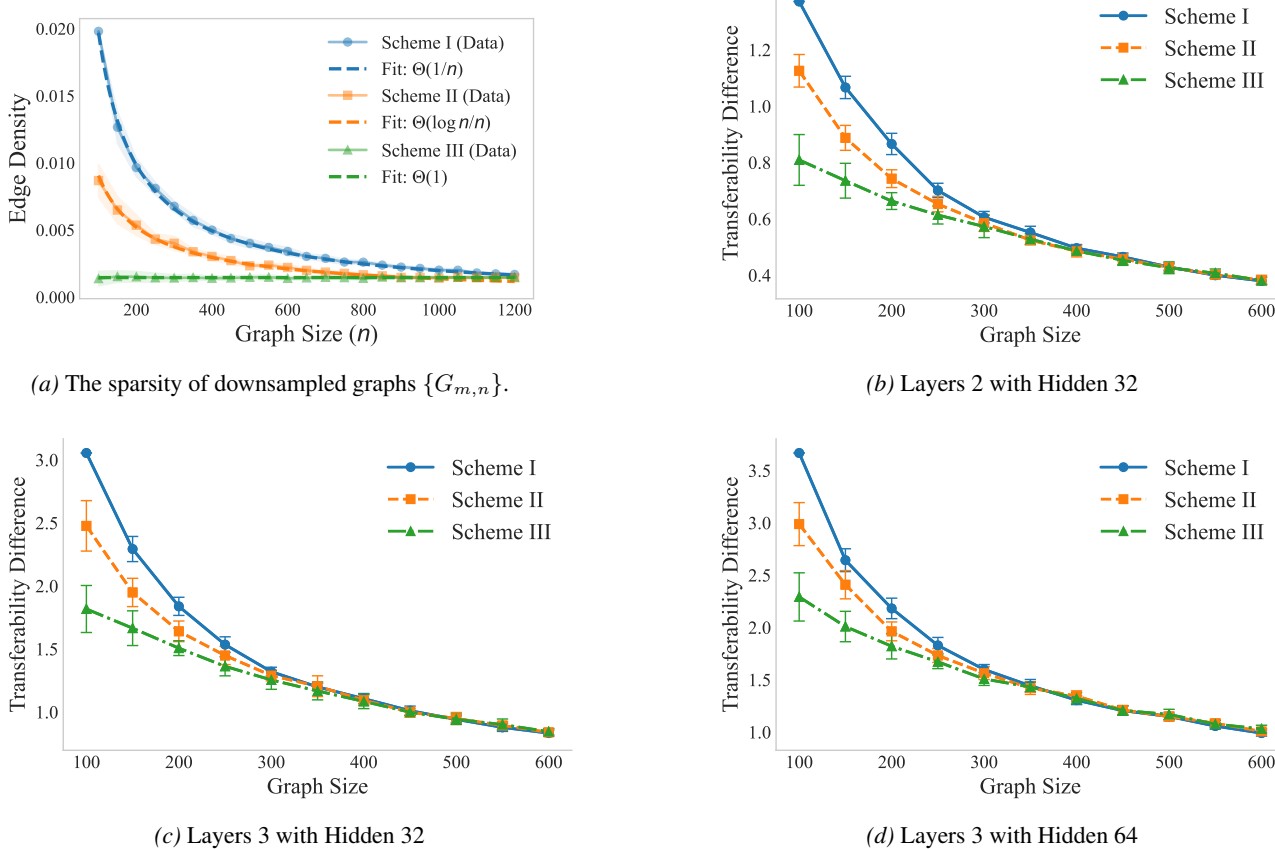

*(a)* The sparsity of downsampled graphs $\{G_{m,n}\}$.

*(b)* Layers 2 with Hidden 32

*(c)* Layers 3 with Hidden 32

*(d)* Layers 3 with Hidden 64

*Figure 5.* GCN transfer errors of sparse graph sequences. Transfer is performed from graphs following sparsity schemes, I: $f(n) = 2/n$; II: $f(n) = 0.2 \log n/n$; III: $f(n) = \Theta(1)$, to the target graph $G_{M,N}$, which corresponds to sparsity functions $3.9/n$ or $1.14 \log n/n$.

## 6. Experiments

To validate the transferability results on sparse graphs, we conduct experiments on the **Cora** dataset (Sen et al., 2008), with additional results on the **Pubmed** and **Ogbn-Arxiv** datasets provided in Appendix A. In the appendix, we also comprehensively evaluate downstream performance (e.g., accuracy, Macro-F1, and cross-entropy loss) and conduct experiments on synthetic geometric random graphs. The source code is available at `https://github.com/ElliotShu/Size_Transferability_of_GCNs_across_Sparsity`.

The Cora graph contains 2708 nodes and has an edge density of 0.00144. We fix $G_{M,N}$ as the full Cora graph, and evaluate the transferability from graphs $G_{m,n}$ with arbitrary sparsity to it. Specifically, we measure the average transfer error over the full sampled graph to analyze transfer errors.

Since the sampled graphs originate from a real-world dataset, the expanding sequence $\{W_m\}$ is approximated by an expanding sequence of subgraphs $\{G_m\}$. We construct $\{G_m\}$ from Cora by adding 50 nodes at each stage in degree order. This ordering is designed to mirror the

construction of the sparse graphon sequence, simulating the expansion from a "dense core" to a "sparse tail" on real finite graphs.

To model different sparsity levels, we define three sparsity schemes: Scheme I: $f(n) = 2/n$; Scheme II: $f(n) = 0.2 \log n/n$; Scheme III: $f(n) = \Theta(1)$ (see Fig. 5a). The transferability under the sparsity of Scheme I is not guaranteed in previous works (Keriven et al., 2020; 2021; Wang et al., 2023; 2024). We set the graph sizes $n$ of $\{G_{m,n}\}$ to be $\{100, 200, ..., 1600\}$. Each sampled $G_{m,n}$ is aligned with a corresponding stage $G_m$, determined by the condition $f(n) \approx e(G_m)$, where $e(G_m)$ denotes the edge density of $G_m$. Each graph $G_{m,n}$ is constructed by randomly selecting $n$ nodes and inheriting their edges and features. Edge densities are shown in Fig. 5a.

The GCN output of the full Cora graph serves as the reference. For each sampled subgraph $G_{m,n}$, we compare the average output error across all nodes against the reference. Unlike the classical graphon setting, which normalizes as $\Phi(\mathbf{S}_n/n, \boldsymbol{x}_n, \mathcal{H})$ (Ruiz et al., 2023), under the stretched form we adopt $\Phi(\mathbf{S}_n/\sqrt{2e_n}, \boldsymbol{x}_n, \mathcal{H})$, where $e_n$ is the number of edges in $G_{m,n}$ (Ji et al., 2024). The difference

arises from the operator impacts of inducing, as shown in Lemma D.2 and Lemma D.5. Each subgraph is sampled 20 times to compute the mean and standard deviation. We further consider GCNs with 2 or 3 layers and 32 or 64 hidden feature dimensions.

Across all sparsity schemes, transferability errors vanish as edge densities decrease and graph sizes increase, as shown in Figure 5. For small or medium graph sizes, sparser graphs exhibit higher edge densities, leading to larger transferability errors. For large graph sizes, sequences under different sparsity converge together, producing similar transferability errors. In Scheme III, where the expected edge density is fixed, increasing the graph size consistently reduces the transfer error. These observations validate our theoretical analysis of edge density and graph size in transferability.

Additionally, GCNs with deeper layers and wider channels generally exhibit larger transfer errors due to the architectural amplification effect, as also illustrated in Figure 5. Strictly speaking, this empirical trend may fluctuate. Our theoretical bounds represent worst-case scenarios; in practice, deep or wide models might learn smaller parameter weights than shallow or narrow ones, potentially altering the exact magnitude of this amplification.

## 7. Conclusion

In this work, we study the transferability of GCNs across varying sparsity levels. Subgraphs are randomly generated from the original graph, resulting in multiple sparsity. By introducing GWCNs as the non-zero limit of GCNs, we establish a transfer error bound over sparse graphs, which is influenced by expected edge density and graph size, factors that jointly determine the multiple sparsity. Experiments on real-world networks validate these theoretical findings.

## Acknowledgement

This work was supported in part by the National Key R&D Program of China under Grant 2024YFE0200700. The computations in this research were performed using the CFFF platform of Fudan University.

## Impact Statement

This work contributes to the foundational theory of Graph Neural Networks, by establishing a theoretical guarantee for size transfer of GCNs across varying sparsity regimes. Our research facilitates more efficient and theoretically-grounded model deployment in complex and large network environments. While this is a fundamental research contribution without direct social risks, it indirectly supports the development of more robust, scalable, and resource-efficient machine learning algorithms.

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

## A. Supplementary Experiments

### A.1. Synthetic Experiment

Following Keriven et al. (2020), we generate geometric random graphs by uniformly sampling latent variables on a 3D manifold. To isolate topological effects, node features are initialized as a constant signal. Edges are independently drawn using a Gaussian similarity kernel $W(x, x') = \alpha(n) \exp(-\|x - x'\|^2/2\sigma^2)$ with a fixed bandwidth $\sigma = 0.15$. We control network sparsity via $\alpha(n) \in \{n^{-1/4}, n^{-1/2}, \log(n)/n, 1/n\}$ and evaluate size transferability across graph sizes $n \in [100, 1000]$ against the continuous GCN (c-GCN) in Keriven et al. (2020).

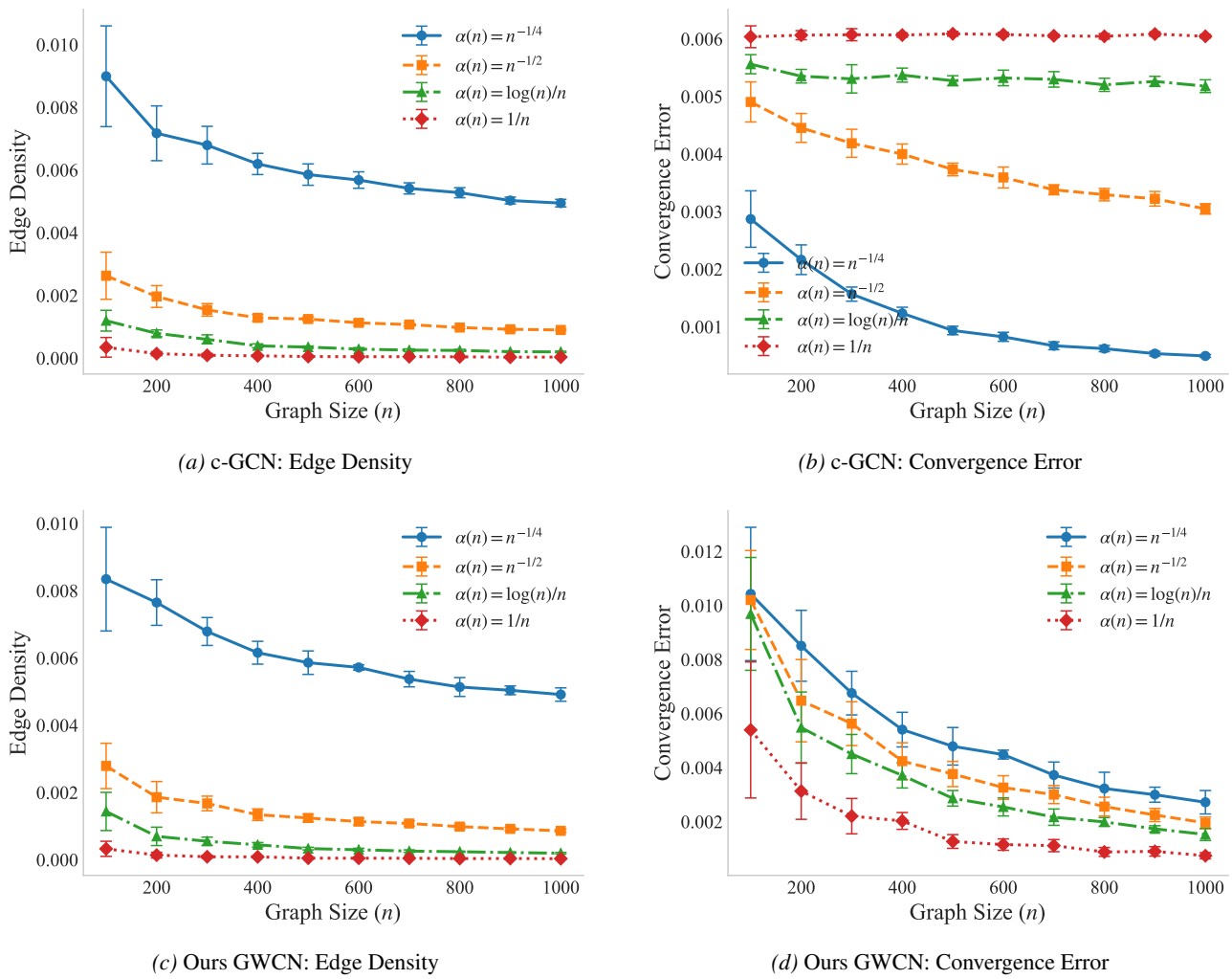

*(a)* c-GCN: Edge Density

*(b)* c-GCN: Convergence Error

*(c)* Ours GWCN: Edge Density

*(d)* Ours GWCN: Convergence Error

*Figure 6.* Comparison of Edge Density scaling and Convergence Error between the baseline c-GCN and our proposed GWCN across different sparsity schemes $\alpha(n)$ as a function of graph size $n$.

As illustrated in Figure 6, panels (a) and (c) confirm identical edge density decay for both models. Crucially, panel (b) reveals that the baseline c-GCN suffers from convergence bottlenecks in sparser graphs, completely stagnating under extreme sparsity ($\alpha(n) = 1/n$). Conversely, panel (d) demonstrates that our GWCN achieves robust convergence; its transfer error steadily decreases even under the most challenging topological conditions ($\alpha(n) = \log(n)/n$ and $1/n$).

## A.2. Cora Dataset: Supplementary Performance Metrics

To validate the empirical behavior beyond just the transfer error, we evaluate a 3-layer GCN with 64 hidden channels on the Cora dataset. We track the scaling properties by varying the sampled graph size $n$ under three sparsity schemes: Scheme I ($2.0/n$), Scheme II ($0.2 \log(n)/n$), and Scheme III (fixed density of $0.0014$). We evaluate four distinct metrics over 20 independent random trials: Transfer Error, Cross-Entropy Loss, Classification Accuracy, and Macro-F1 Score. The results are illustrated in Figure 7.

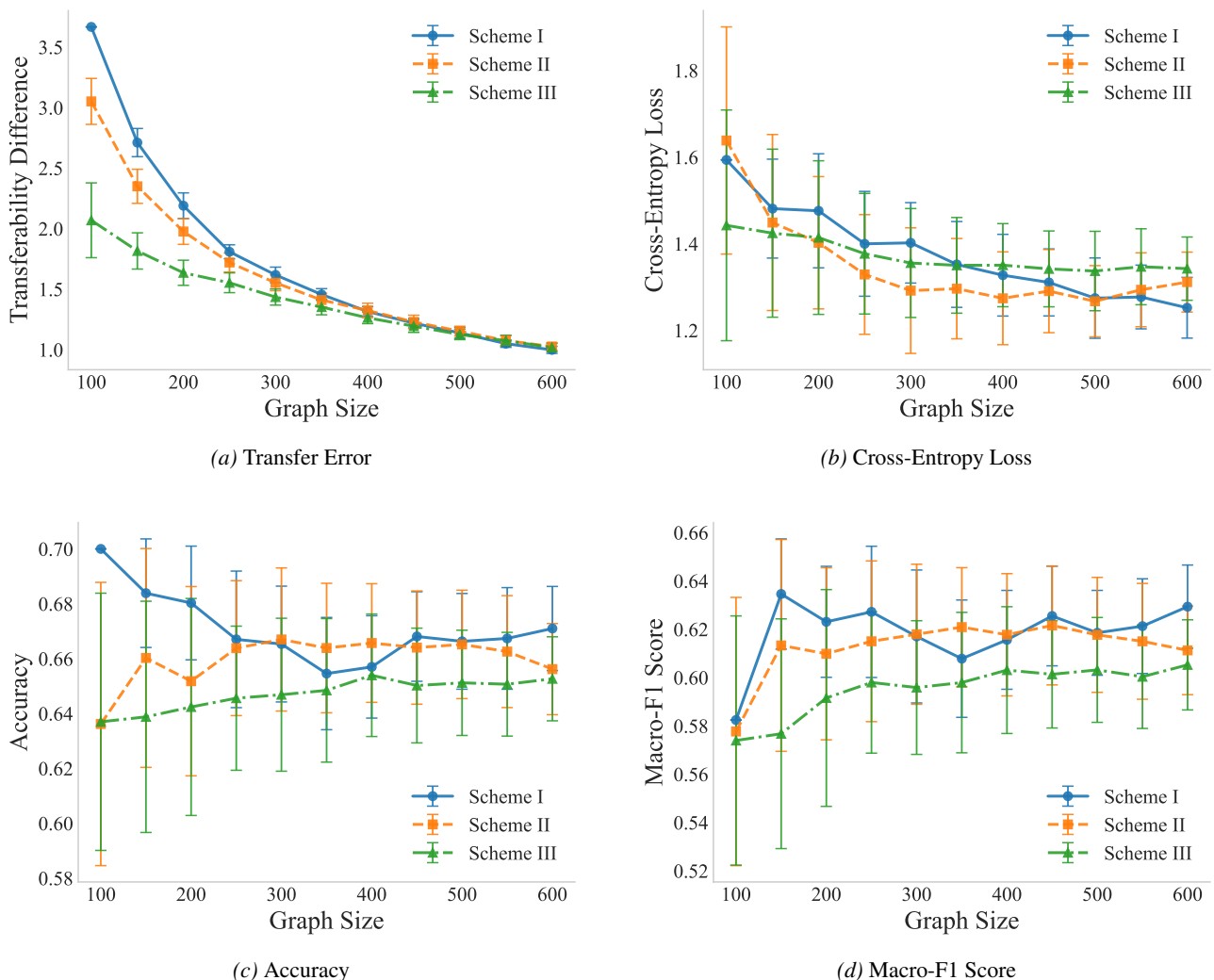

*Figure 7.* Comprehensive metric evaluation on the Cora dataset using a 3-layer GCN with 64 hidden channels. The solid lines display the mean values computed across 20 trials, while the error bars denote the standard deviation, illustrating the variance and stability at each subgraph scale.

As shown in Figure 7, the scaling trends reveal that as the graph size grows, not only does the transfer error converge toward zero, but all other downstream metrics also demonstrate convergence with progressively shrinking variance. Unlike continuous transfer error, classification metrics possess a margin of tolerance. As long as the transfer error remains within decision boundaries, the downstream predictions remain unchanged. Therefore, the transfer error decreases smoothly while accuracy and F1 score remain highly stable. This observation verifies that the "bounded error tending to zero" established in Theorem 5.1 translates directly into robust downstream task performance.

## A.3. Pubmed Dataset

The Pubmed graph consists of 19,717 nodes with an edge density of 0.000228, mapping to theoretical sparsity functions of $4.5/n$ or $1.05 \log n/n$. We designate the entire graph as the target benchmark $G_{M,N}$ to systematically evaluate the transfer error from the downsampled subgraphs $G_{m,n}$.

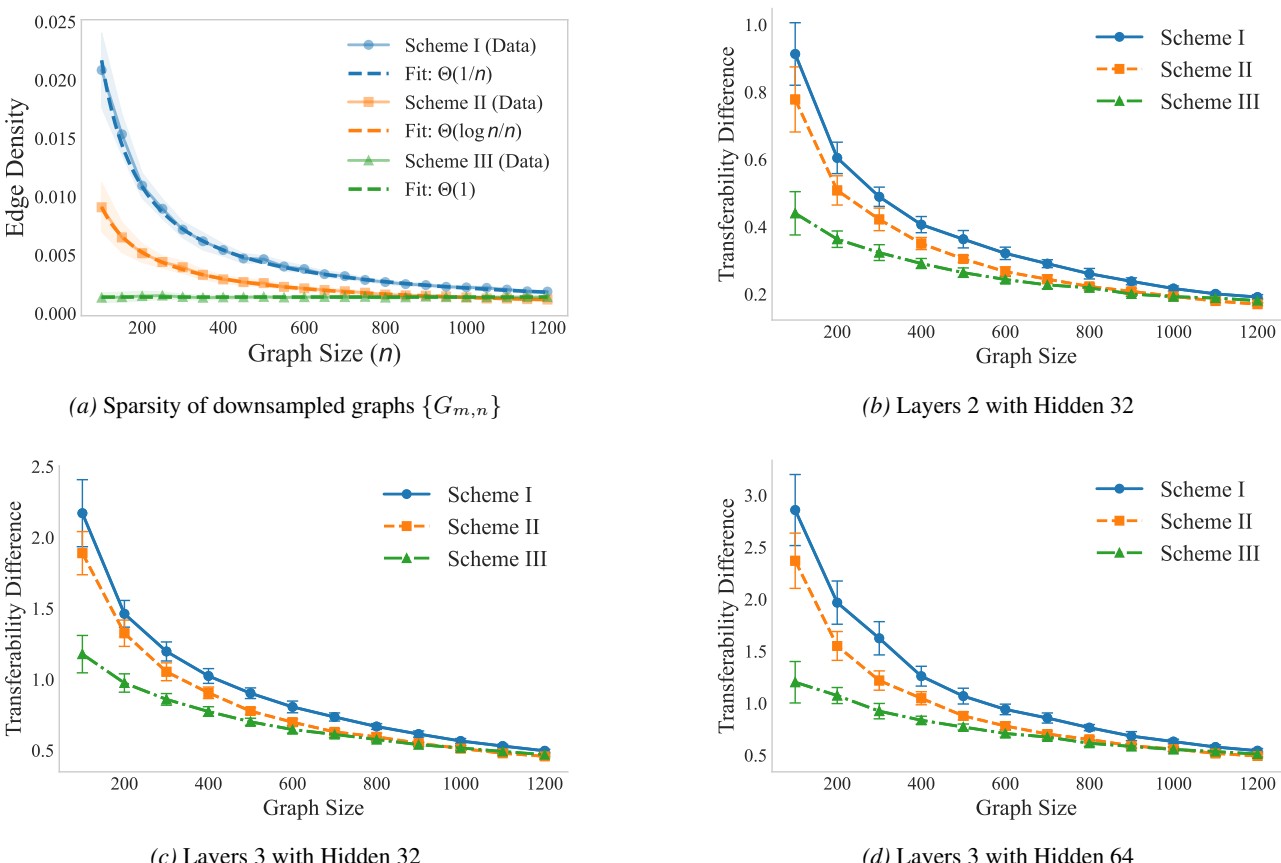

*(a)* Sparsity of downsampled graphs $\{G_{m,n}\}$

*(b)* Layers 2 with Hidden 32

*(c)* Layers 3 with Hidden 32

*(d)* Layers 3 with Hidden 64

*Figure 8.* GCN transfer errors of sparse graph sequences on Pubmed. Transfer is performed from graphs following sparsity schemes (I: $f(n) = 2.2/n$; II: $f(n) = 0.2 \log n/n$; III: $f(n) = \Theta(1)$) to the target graph $G_{M,N}$, which aligns with sparsity functions $4.5/n$ or $1.05 \log n/n$.

To model varying sparsity levels during the transition, we introduce three distinct sampling schemes: Scheme I ($f(n) = 2.2/n$), Scheme II ($f(n) = 0.2 \log n/n$), and Scheme III ($f(n) = \Theta(1)$), as depicted in Figure 8(a). The empirical results consistently demonstrate our core conclusion: increasing the subgraph size while maintaining a lower edge density effectively minimizes the transfer error.

### A.4. Ogbn-Arxiv Dataset

To verify our theoretical claims on a substantially larger scale, we extend the evaluation to the Ogbn-Arxiv dataset. This citation network contains 169,343 nodes with an even lower edge density of $0.000081$, bounded by sparsity functions of $13.7/n$ or $1.14 \log n/n$.

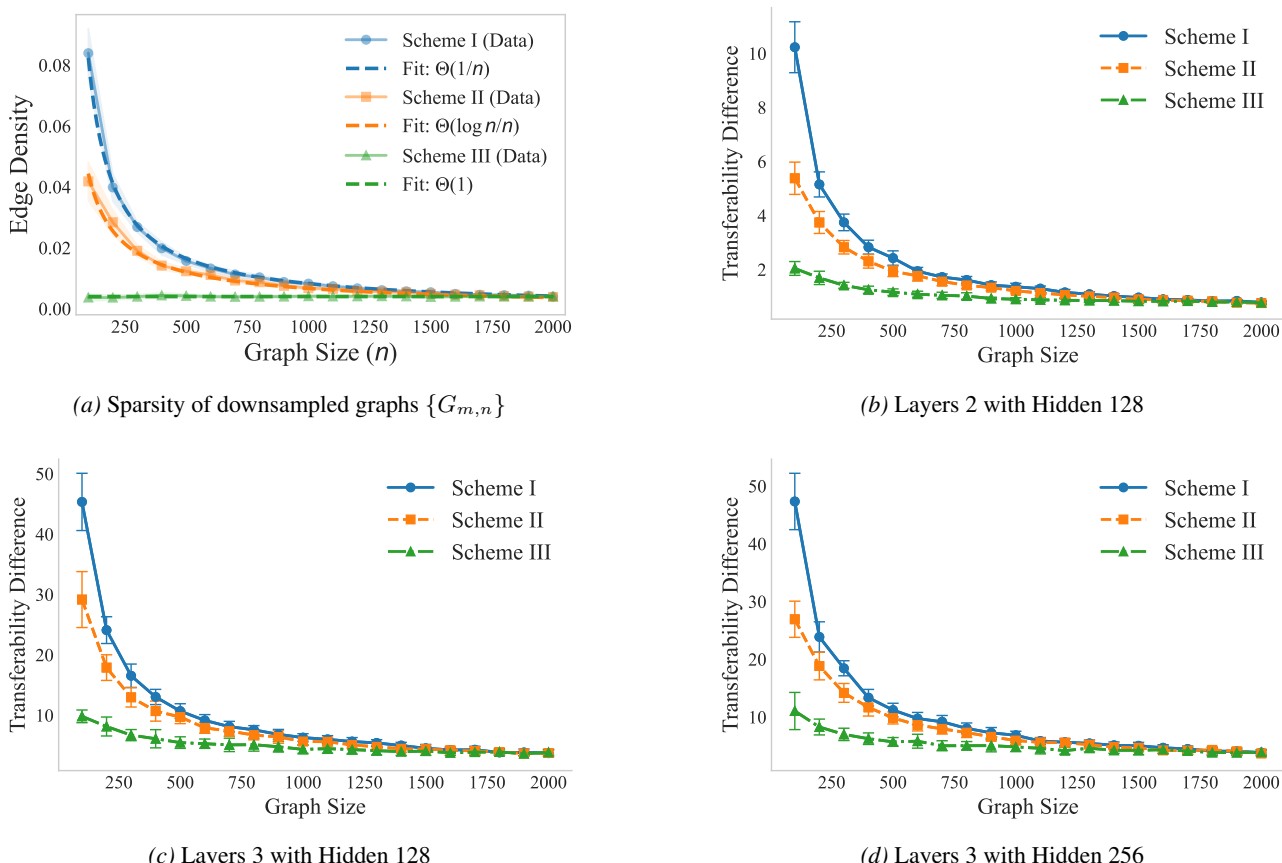

*(a)* Sparsity of downsampled graphs $\{G_{m,n}\}$

*(b)* Layers 2 with Hidden 128

*(c)* Layers 3 with Hidden 128

*(d)* Layers 3 with Hidden 256

*Figure 9.* GCN transfer errors of sparse graph sequences on Ogbn-Arxiv. Transfer is performed from graphs following scaled sparsity schemes (I: $f(n) = 10/n$; II: $f(n) = \log n/n$; III: $f(n) = \Theta(1)$) to the target graph $G_{M,N}$, which aligns with sparsity functions $13.7/n$ or $1.14 \log n/n$.

Following the established protocol, the full graph serves as the transfer target $G_{M,N}$. Here, the sparsity configurations are scaled to reflect the network's extreme sparsity: Scheme I ($f(n) = 10/n$), Scheme II ($f(n) = \log n/n$), and Scheme III ($f(n) = \Theta(1)$). As shown in Figure 9, the results on this large-scale network echo our previous findings, further corroborating that superior transferability is achieved under conditions of larger graph scales and sparser connections.

# B. Appendix: Generalized Concepts for Analysis

In some graphon analyses, new functions are constructed, such as $\mathcal{W} = W_0 - W_1$, the value range of which exceeds the probability range $[0, 1]$ in Definition 3.1. To make our proof more concise and clear, we introduce several related generalized concepts.

**Definition B.1** (Value-unbounded Graphon). By just extending the value range of graphon $[0, 1]$ into $\mathbb{R}$, the value-unbounded graphon is denoted by $\mathcal{W} : \mathbb{R}_+^2 \to \mathbb{R}$.

Similarly, we consider not only the adjacency matrix $\mathbf{S}_n \in \{0, 1\}^{n \times n}$ consisting of binary entries, but also symmetric matrices with wider range of values, denoted by $\mathcal{S}_n \in \mathbb{R}^{n \times n}$, whose induced functional form is given by

$$\overline{\mathcal{W}}_n(u \in \mathbb{I}_i, v \in \mathbb{I}_j) = \mathcal{S}_n(i, j), \tag{12}$$

which is only supported over $[0, 1]^2$.

In Definition 3.2, stretching is defined using a specific coefficient $s = \sqrt{|W|_1}$, which is associated with graphon $W$ to be stretched. Accordingly, we introduce a more general form of stretching, in which the coefficient is not restricted to any specific graphon, denoted by $s_\forall \in (0, 1]$.

**Definition B.2** (Non-specific Stretching). For value-unbounded graphon $\mathcal{W}(u, v)$, the stretched graphon $\mathcal{W}^{s_\forall}$ and the stretched graphon signal is defined by,

$$\begin{aligned}
\mathcal{W}^{s_\forall}(u, v) &= \mathcal{W}(s_\forall u, s_\forall v), \\
X^{s_\forall}(u) &= X(s_\forall u),
\end{aligned} \tag{13}$$

where $s_\forall \in (0, 1]$ is non-specific stretching coefficient.

The non-specific stretching of classical graphon and signal is similar to Eq.(5), by replacing the specific $s$ with non-specific $s_\forall$.

# C. Convergence of Sparse Graphon Sequence: Proof

We write down the constants or values approaching constants generated in the following proofs of this section as follows, where we require $k$ to satisfy $k' = (k - 3)/2 > 0$:

$$\begin{aligned}
C_{k,m} &= \frac{2C_k}{(k-1)^2}, \\
C_{k',W} &= \frac{C_{k,m}}{s_R} + A_W \sqrt{\frac{21}{8} \frac{C_{k,m}}{s_R^3}} + \frac{\sqrt{2C_{k,m}}}{s_R^2}, \\
C_{k',X} &= A_X \frac{2}{\sqrt{27}} \frac{C_{k,m}}{s_R^2} + \sqrt{\frac{4 \max(X) C_{k,m}}{3 s_R^3}}.
\end{aligned} \tag{14}$$

The equivalence between $W_m^s$ and $W_m'^s$, and the equivalence between $X_m^s$ and $X_m'^s$ have been discribed in Eq.(15).

$$\begin{aligned}
W_m^s(u, v) &= W_m\left(\frac{s_m'}{t_m}u, \frac{s_m'}{t_m}v\right) = W_m'(s_m'u, s_m'v) = W_m'^s(u, v), \\
X_m^s(u) &= X_m\left(\frac{s_m'}{t_m}u\right) = X_m'(s_m'u) = X_m'^s(u).
\end{aligned} \tag{15}$$

Therefore, the convergence of $W_m^s$ and $X_m^s$ is equivalent to that of $W_m'^s$ and $X_m'^s$. Therefore, in the following discussion, we focus directly on the convergence of $W_m'^s$ and $X_m'^s$.

## C.1. Stretching Coefficients of Sparse Graphon

We denote the stretching coefficients of $W_m'^s$ and $W_{\mathbb{R}_+}^s$ by $s_m'$ and $s_R$. According to the definition of stretching and the truncation process, we have

$$\left| s_R^2 - s_m'^2 \right| = \left| \|W_{\mathbb{R}_+}\|_1 - \|W_m'\|_1 \right| = \|W_{\mathbb{R}_+}\|_{L^1(\mathbb{R}_+^2/[0, t_m]^2)}. \tag{16}$$

Considering that $W_{\mathbb{R}_+}$ belongs to the Schwartz space, i.e., for any $k > 0$, there exits a constant $C_k$ such that

$$W_{\mathbb{R}_+}(u, v) \leq \frac{C_k}{(u+1)^k (v+1)^k}. \tag{17}$$

Based on it, we derive the upper bound of $\left| s_R^2 - s_m^{'2} \right|$:

$$
\begin{aligned}
\left| s_R^2 - s_m^{'2} \right| &\leq \left\| \frac{C_k}{(u+1)^k (v+1)^k} \right\|_{L^1(\mathbb{R}_+^2 / [0, t_m]^2)} \\
&\leq \int_{u \geq t_m} \int_{\mathbb{R}_+} \frac{C_k}{(u+1)^k (v+1)^k} dv du + \int_{\mathbb{R}_+} \int_{v \geq t_m} \frac{C_k}{(u+1)^k (v+1)^k} dv du \\
&= 2 \int_{u \geq t_m} \int_{\mathbb{R}_+} \frac{C_k}{(u+1)^k (v+1)^k} dv du.
\end{aligned}
\tag{18}
$$

By setting $k > 1$, the integral in the above expression can be evaluated, yielding

$$\left| s_R^2 - s_m^{'2} \right| \leq \frac{2C_k}{(k-1)^2} (1 + t_m)^{-k+1} \leq \frac{2C_k}{(k-1)^2} \frac{1}{t_m^{k-1}}. \tag{19}$$

### C.2. Stretched Graphon of Sparse Graphon

For clarity, we denote the stretched forms of $W_{\mathbb{R}_+}$ and $W_m'$ by $W_{\mathbb{R}_+}^{s_R}$ and $W_m^{'s_m'}$, so that the stretch defined with specific coefficients can be distinguished from that defined with non-specific coefficients. In the proof, we introduce an intermediate term $W_m^{'s_R}$ according to Definition B.2. By the triangle inequality, we have:

$$
\begin{aligned}
\left\| W_{\mathbb{R}_+}^{s} - W_m^{s} \right\|_{2,2} &\leq \left\| W_{\mathbb{R}_+}^{s_R} - W_m^{'s_R} \right\|_{2,2} + \left\| W_m^{'s_R} - W_m^{'s_m'} \right\|_{2,2} \\
&\leq \left\| W_{\mathbb{R}_+}^{s_R} - W_m^{'s_R} \right\|_2 + \left\| W_m^{'s_R} - W_m^{'s_m'} \right\|_2,
\end{aligned}
\tag{20}
$$

the transition from the operator norm to the $L^2$-norm relies on Lemma 1 in Ji et al. (2024).

For the first part, since $W_m'$ is directly obtained from $W_{\mathbb{R}_+}$ by truncation, and both share the same stretching factor according to Definition B.2, we have

$$W_m^{'s_R}(u, v) = W_{\mathbb{R}_+}^{s_R}(u, v), \quad (u, v) \in [0, t_m / s_R]^2.$$

Hence, noting that $W_{\mathbb{R}_+} \leq 1$, this part reduces to

$$
\begin{aligned}
\left\| W_{\mathbb{R}_+}^{s_R} - W_m^{'s_R} \right\|_2 &= \frac{1}{s_R} \left\| W_{\mathbb{R}_+} \right\|_{L^2(\mathbb{R}_+^2 / [0, t_m]^2)} \\
&\leq \frac{1}{s_R} \left\| W_{\mathbb{R}_+} \right\|_{L^1(\mathbb{R}_+^2 / [0, t_m]^2)} = \frac{1}{s_R} \left| s_R^2 - s_m^{'2} \right|.
\end{aligned}
\tag{21}
$$

For the second part, it should be noted that $W_m^{'s_R}$ and $W_m^{'s_m'}$ are supported on different domains, namely $[0, t_m / s_R]^2$ and $[0, t_m / s_m']^2$, respectively. Consequently, when evaluating their discrepancy in terms of the $L^2$-norm, the integration has to be considered separately over the corresponding regions:

$$
\begin{aligned}
\left\| W_m^{'s_R} - W_m^{'s_m'} \right\|_2^2 &= \int \int_{\mathbb{R}_+^2} \left( W_m^{'s_R}(u, v) - W_m^{'s_m'}(u, v) \right)^2 du dv \\
&= \underbrace{\left\| W_m^{'s_R} - W_m^{'s_m'} \right\|_{L^2([0, t_m / s_R]^2)}^2}_{\text{(i)}} + \underbrace{\left\| W_m^{'s_m'} \right\|_{L^2([0, t_m / s_m']^2 / [0, t_m / s_R]^2)}^2}_{\text{(ii)}},
\end{aligned}
\tag{22}
$$

where the second term accounts for the mismatch of domains.

By expanding the integral of the first term in the above expression in detail, we obtain

$$
\underbrace{\left\| W_m^{'s_R} - W_m^{'s'_m} \right\|_{L^2([0,t_m/s_R]^2)}^2}_{(i)} = \int\int_{[0,\frac{t_m}{s_R}]^2} \left( W_{\mathbb{R}_+}(s_R u, s_R v) - W_{\mathbb{R}_+}(s'_m u, s'_m v) \right)^2 du\,dv
$$

$$
\leq A_W^2 \int\int_{[0,\frac{t_m}{s_R}]^2} \left( |s_R u - s'_m u| + |s_R v - s'_m v| \right)^2 du\,dv \tag{23}
$$

$$
= A_W^2 \frac{7 t_m^4}{6 s_R^4} (s_R - s'_m)^2,
$$

where the inequality is derived by the Lipschitz property of $W_{\mathbb{R}_+}$.

For the second term (ii), since the maximum value of the function is 1, the integral result is obtained by evaluating the mismatch area,

$$
\underbrace{\left\| W_m^{'s'_m} \right\|_{L^2([0,t_m/s'_m]^2/[0,t_m/s_R]^2)}^2}_{(ii)} \leq \left( \frac{t_m}{s'_m} \right)^2 - \left( \frac{t_m}{s_R} \right)^2 = \frac{t_m^2}{s_m'^2 s_R^2} \left( s_R^2 - s_m'^2 \right). \tag{24}
$$

Assume $t_m$ is sufficiently large such that $\|W'_m\|_1 \geq \|W_{\mathbb{R}_+}\|_1/2$, then the above inequalities becomes

$$
\underbrace{\left\| W_m^{'s_R} - W_m^{'s'_m} \right\|_{L^2([0,t_m/s_R]^2)}^2}_{(i)} \leq A_W^2 \frac{63 t_m^4}{24 s_R^6} (s_R^2 - s_m'^2)^2,
$$

$$
\underbrace{\left\| W_m^{'s'_m} \right\|_{L^2([0,t_m/s'_m]^2/[0,t_m/s_R]^2)}^2}_{(ii)} \leq \frac{2 t_m^2}{s_R^4} \left( s_R^2 - s_m'^2 \right). \tag{25}
$$

Substituting inequality 19 into the above expression 25, in order to offset the effect of $t_m$ in the numerator, we impose the condition $k - 1 > 2$. By further combining this with the previous inequality 21, we obtain

$$
\left\| W_{\mathbb{R}_+}^s - W_m^s \right\|_{2,2} \leq \frac{1}{s_R} \left| s_R^2 - s_m'^2 \right| + A_W \sqrt{\frac{63}{24 s_R^6}} t_m^2 \left| s_R^2 - s_m'^2 \right| + \sqrt{\frac{2}{s_R^4}} t_m \sqrt{\left| s_R^2 - s_m'^2 \right|}
$$

$$
\leq \frac{C_{k,m}}{s_R} \frac{1}{t_m^{k-1}} + A_W \sqrt{\frac{63}{24 s_R^6}} \frac{C_{k,m}}{t_m^{k-3}} + \sqrt{\frac{2}{s_R^4}} \frac{\sqrt{C_{k,m}}}{t_m^{(k-3)/2}} \tag{26}
$$

where $C_{k,m} = \frac{2 C_k}{(k-1)^2}$ denotes a general form of the constant in the inequality $\left| s_R^2 - s_m'^2 \right| \leq \frac{2 C_k}{(k-1)^2} \frac{1}{t_m^{k-1}}$ $(k > 1)$. To simplify the presentation, we replace $(k-3)/2$ with $k' = (k-3)/2$, and by applying contraction $t_m^{-(k-1)} \leq t_m^{-(k-3)} \leq t_m^{-(k-3)/2}$, the above result is transformed into

$$
\left\| W_{\mathbb{R}_+}^s - W_m^s \right\|_{2,2} \leq \frac{C_{k',W}}{t_m^{k'}}, \tag{27}
$$

where $k' > 0$ and $C_{k',W} = \frac{C_{k,m}}{s_R} + A_W \sqrt{\frac{63}{24 s_R^6}} C_{k,m} + \frac{\sqrt{2 C_{k,m}}}{s_R^2}$.

### C.3. Stretched Signal of Sparse Graphon

For clarify, We denote the stretched forms of $X_{\mathbb{R}_+}$ and $X'_m$ by $X_{\mathbb{R}_+}^{s_R}$ and $X_m^{'s'_m}$. Based on Definition B.2, we introduce an intermediate term $X_m^{'s_R}$. By the triangle inequality, we have:

$$
\left\| X_{\mathbb{R}_+}^s - X_m^s \right\|_2 \leq \left\| X_{\mathbb{R}_+}^{s_R} - X_m^{'s_R} \right\|_2 + \left\| X_m^{'s_R} - X_m^{'s'_m} \right\|_2
$$

$$
\leq \left\| X_{\mathbb{R}_+}^{s_R} - X_m^{'s_R} \right\|_2 + \left\| X_m^{'s_R} - X_m^{'s'_m} \right\|_2. \tag{28}
$$

For the first part, since $X'_m$ is directly obtained from $X_{\mathbb{R}_+}$ by truncation, and both share the same stretching coefficient, we have

$$X'^{s_R}_m(u) = X^{s_R}_{\mathbb{R}_+}(u), \quad u \in [0, t_m/s_R].$$

Hence, this part reduces to

$$\left\| X^{s_R}_{\mathbb{R}_+} - X'^{s_R}_m \right\|_2 = \frac{1}{\sqrt{s_R}} \left\| X_{\mathbb{R}_+} \right\|_{L^2(\mathbb{R}_+/[0,t_m])}. \tag{29}$$

For the second part, $X'^{s_R}_m$ and $X'^{s'_m}_m$ are supported on $[0, t_m/s_R]$ and $[0, t_m/s'_m]$ differently. Consequently, it's considered separately over the corresponding regions:

$$\left\| X'^{s_R}_m - X'^{s'_m}_m \right\|_2^2 = \int_{\mathbb{R}_+} \left( X'^{s_R}_m(u) - X'^{s'_m}_m(u) \right)^2 du$$
$$= \underbrace{\left\| X'^{s_R}_m - X'^{s'_m}_m \right\|^2_{L^2([0,t_m/s_R])}}_{\text{(i)}} + \underbrace{\left\| X'^{s'_m}_m \right\|^2_{L^2([t_m/s'_m, t_m/s_R])}}_{\text{(ii)}}, \tag{30}$$

where the second term accounts for the mismatch of domains. Expanding the integral of the first term in detail, we obtain

$$\underbrace{\left\| X'^{s_R}_m - X'^{s'_m}_m \right\|^2_{L^2([0,t_m/s_R])}}_{\text{(i)}} = \int_{[0,\frac{t_m}{s_R}]} \left( X_{\mathbb{R}_+}(s_R u) - X_{\mathbb{R}_+}(s'_m u) \right)^2 du$$
$$\leq A_X^2 \int_{[0,\frac{t_m}{s_R}]} (s_R u - s'_m u)^2 du \tag{31}$$
$$= A_X^2 \frac{t_m^2}{3s_R^2} (s_R - s'_m)^2,$$

where the inequality is derived by the Lipschitz property of $X_{\mathbb{R}_+}$. For the second term (ii), we denote the maximum value of $X_{\mathbb{R}_+}$ by $\max(X_{\mathbb{R}_+})$, then the integral result is obtained by evaluating the mismatch area,

$$\underbrace{\left\| X'^{s'_m}_m \right\|^2_{L^2([t_m/s'_m, t_m/s_R])}}_{\text{(ii)}} \leq \max(X) \left( \frac{t_m}{s_R} - \frac{t_m}{s'_m} \right) = \frac{\max(X) t_m}{s'_m s_R} (s_R - s'_m). \tag{32}$$

When $t_m$ is sufficiently large such that $s'^2_m \geq s_R^2/2$, then the above inequalities becomes

$$\underbrace{\left\| X'^{s_R}_m - X'^{s'_m}_m \right\|^2_{L^2([0,t_m/s_R])}}_{\text{(i)}} \leq A_X^2 \frac{4t_m^2}{27s_R^4} (s_R^2 - s'^2_m)^2,$$

$$\underbrace{\left\| X'^{s'_m}_m \right\|^2_{L^2([0,t_m/s'_m]/[0,t_m/s_R])}}_{\text{(ii)}} \leq \frac{4\max(X) t_m}{3s_R^3} (s_R^2 - s'^2_m). \tag{33}$$

Similarly, to offset the effect of $t_m$ in the numerator, we impose the condition $k - 1 > 1$ of the inequality 19. Combining this with the previous results, we obtain

$$\left\| X^s_{\mathbb{R}_+} - X^s_m \right\|_2 \leq \frac{1}{\sqrt{s_R}} \left\| X_{\mathbb{R}_+} \right\|_{L^2(\mathbb{R}_+/[0,t_m])}$$
$$+ A_X \sqrt{\frac{4}{27s_R^4}} t_m \left| s_R^2 - s'^2_m \right| + \sqrt{\frac{4\max(X)}{3s_R^3}} \sqrt{t_m \left| s_R^2 - s'^2_m \right|} \tag{34}$$
$$\leq \frac{\left\| X_{\mathbb{R}_+} \right\|_{L^2([t_m,+\infty))}}{\sqrt{s_R}} + A_X \sqrt{\frac{4}{27s_R^4}} \frac{C_{k,m}}{t_m^{k-2}} + \sqrt{\frac{4\max(X)}{3s_R^3}} \frac{\sqrt{C_{k,m}}}{t_m^{(k-2)/2}}.$$

To simplify the presentation, we replace $(k-2)/2$ with $k' = (k-2)/2$, and by applying contraction $t_m^{-(k-2)} \leq t_m^{-(k-2)/2}$, the above result is transformed into

$$\left\| X_{\mathbb{R}_+}^{\mathfrak{s}} - X_m^{\mathfrak{s}} \right\|_2 \leq \frac{C_{k',X}}{t_m^{k'}} + \frac{\|X_{\mathbb{R}_+}\|_{L^2([t_m,+\infty))}}{\sqrt{s_R}}, \tag{35}$$

where $k' > 0$ and $C_{k',X} = A_X \sqrt{\frac{4}{27s_R^4}} C_{k,m} + \sqrt{\frac{4\max(X)C_{k,m}}{3s_R^3}}$.

## D. Convergence of Random Graph Sequence: Proof

At each stage $m$, the random graph sequence $\{G_{m,n}\}$ has its associated limit $W_m$. This convergence holds in both the classical form and the stretched form. Moreover, convergence in the stretched form is implied by convergence in the classical form. Their notation is summarized in the following Table 2.

*Table 2.* Notation for convergence errors in random graph sequence.

| Graphon form | Operator difference | Signal difference |
|---|---|---|
| Classical form | $\Delta T_{W_n} : \|W_m - \overline{W}_{m,n}\|_{2,2}$ | $\Delta X_n : \|X_n - \overline{X}_{m,n}\|_2$ |
| Stretched form | $\Delta T_{W_n^{\mathfrak{s}}} : \|W_m^{\mathfrak{s}} - \overline{W}_{m,n}^{\mathfrak{s}}\|_{2,2}$ | $\Delta X_n^{\mathfrak{s}} : \|X_m^{\mathfrak{s}} - \overline{X}_{m,n}^{\mathfrak{s}}\|_2$ |

**To simplify notation**, throughout this part of the proof we write $W, \overline{W}_n$ instead of the full symbols $W_m, \overline{W}_{m,n}$, and similarly $X, \overline{X}_n$ for $X_m, \overline{X}_{m,n}$. The rationale is that when a graph sequence converges to the graphon and graphon signal, the graphon and signal is considered invariant.

We also write down the constants or values approaching constants generated in the following proofs of this section as follows, where $n$ is sufficiently large such that $\Delta T_{W_n} \leq s_m^2/2$:

$$C_W = \frac{C_\nu}{t_m} + \frac{2A_W}{\sqrt{6}\epsilon_u}, \quad C_X = \frac{2A_X}{\sqrt{6}\epsilon_u}.$$

$$C_{W,1} = \frac{2\sqrt{7}A_W t_m}{\sqrt{3}s_m^3(1+1/\sqrt{2})}, \quad C_{W,2} = \frac{2}{s_m^2\sqrt{(1+1/\sqrt{2})/\sqrt{2}}}.$$

$$C_{X,1} = \frac{2^{5/4}A_X t_m}{\sqrt{3}s_m^{5/2}(1+1/\sqrt{2})}, \quad C_{X,2} = \frac{2^{5/8}\max(|X|)}{\sqrt{s_m^{5/2}/\sqrt{2}(1+1/\sqrt{2})}} \tag{36}$$

$$C_{\mathfrak{s}} := \max\{\frac{\sqrt{2}}{s_m}, \frac{2^{\frac{1}{4}}}{\sqrt{s_m}}\},$$

$$C_{W,\mathfrak{s}} = C_{W,1}\Delta T_{W_n}^{1/2} + C_{W,2} \to C_{W,2}, \quad C_{X,\mathfrak{s}} = C_{X,1}\Delta T_{W_n}^{1/2} + C_{X,2} \to C_{X,2}.$$

$C_{\mathfrak{s}} := \max\{\frac{\sqrt{2}}{s}, \frac{2^{\frac{1}{4}}}{\sqrt{s}}\}$ is constant about $W$, $C_{W,\mathfrak{s}} = C_{W,1}\Delta T_{W_n}^{1/2} + C_{W,2}$ and $C_{X,\mathfrak{s}} = C_{X,1}\Delta T_{W_n}^{1/2} + C_{X,2}$ converge to constants $C_{W,2}$ and $C_{X,2}$ as $n$ increases.

### D.1. Classical Form: Proof of Lemma D.1

**Lemma D.1.** *(Convergence to Classical Graphon)*

*Consider the classical graphon and signal $(W_m, X_m)$, and the induced graphon and signal $(\overline{W}_{m,n}, \overline{X}_{m,n})$ sampled from them. For any $\nu > 0$, assume $n$ is large enough such that $n^{-\nu} \leq \epsilon_b$, there exists a constant $C_\nu$ such that*

$$\Delta T_{W_n} \leq \frac{C_W}{\sqrt{n}}t_m, \ \Delta X_n \leq \frac{C_X}{\sqrt{n}}t_m \tag{37}$$

*with probability at least $(1-\epsilon_u)(1-\epsilon_b)$, where $\epsilon_u, \epsilon_b \in (0,1)$ are related to the Uniform sampling and Bernoulli sampling, $C_W = \frac{C_\nu}{t_m} + \frac{2A_W}{\sqrt{6}\epsilon_u}$ and $C_X = \frac{2A_X}{\sqrt{6}\epsilon_u}$ are constants about $W_m$ and $X_m$.*

Under the classical graphon form, as the graph size increases, the random graphs converge to the limit $W_m, X_m$ at the rate of $O(1/n^{1/2})$. However, when $W_m$ diminishes due to sparsity, the full random graph sequence converges to the trivial zero limit.

For the graph topological sampling process, modeled as $W \sim \overline{P}_n \sim \overline{W}_n$, where $\overline{P}_n$ is the induced graphon of the probability matrix. The procedure involves two steps: first, a probability matrix $\mathbf{P}_n$ is sampled from $W$; second, an adjacency matrix $\mathbf{S}_n$ is sampled from the probability matrix. In the graph signal sampling process, where $X \sim \overline{X}_n$, the discrepancy arises due to uniform sampling. Accordingly, the difference can be decomposed into two components:

- Uniform Sampling: $\left\| X - \overline{X}_n \right\|_2$, $\left\| W - \overline{P}_n \right\|_{2,2}$.

- Bernoulli Sampling: $\left\| \overline{P}_n - \overline{W}_n \right\|_{2,2}$.

As the graph size $n$ increases, the graph and graph signal converge to the corresponding graphon and graphon signal. As we do not consider the effect of different vertex permutations of the same graph, we assume that the vertices are ordered according to their latent features, such that $u_1 \le u_2 \le ... \le u_n$. To provide a clear exposition, we decompose the conclusion of Lemma D.1 into two components: convergence of graph signals D.1.2 and convergence of graph structures D.1.3.

Notably, $W_{\mathbb{R}_+}$ and $X_{\mathbb{R}_+}$ satisfy the $A_W$-Lipschitz and $A_X$-Lipschitz properties. In contrast, $W_m$ and $X_m$, being derived through stretching, satisfy the corresponding $t_m A_W$-Lipschitz and $t_m A_X$-Lipschitz properties.

### D.1.1. NORMALIZATION OF CLASSICAL INDUCING

**Lemma D.2.** *Let $(\overline{W}_n, \overline{X}_n)$ be the induced form of value-unbounded graph matrix $\mathcal{S}_n$ ($\mathcal{S}_n \in \mathbb{R}^{n \times n}$) and node features $\boldsymbol{x}_n$. Then it holds*

$$\left\| \overline{X}_n \right\|_2 = \frac{1}{\sqrt{n}} \| \boldsymbol{x}_n \|_2,$$

$$\left\| \overline{W}_n \right\|_{2,2} = \frac{1}{n} \left\| \mathcal{S}_n \right\|_{2,2}. \tag{38}$$

*Proof.* For the $L^2$-norm property between $\overline{X}_n$ and $\boldsymbol{x}_n$,

$$\begin{aligned}
\left\| \overline{X}_n \right\|_2^2 &= \int_{[0,1]} \overline{X}_n^2(u) du = \sum_{i=1}^n \frac{1}{n} \overline{X}_n^2(u \in \mathbb{I}_i) \\
&= \frac{1}{n} \sum_{i=1}^n \boldsymbol{x}_n^2(i) = \frac{1}{n} \| \boldsymbol{x}_n \|_2^2.
\end{aligned} \tag{39}$$

For the operator norm property, according to the definition of $\left\| \overline{W}_n \right\|_{2,2}$, we need to calculate the supremum of $\frac{\left\| T_{\overline{W}_n} X \right\|_2}{\|X\|_2}$ for all $X \in L^2([0,1])$.

$$\begin{aligned}
\frac{\left\| T_{\overline{W}_n} X \right\|_2^2}{\|X\|_2^2} &= \frac{\int_{[0,1]} \left( \int_{[0,1]} \overline{W}_n(u,v) X(u) du \right)^2 dv}{\|X\|_2^2} \\
&= \frac{\int_{[0,1]} \left( \sum_{i=1}^n \overline{W}_n(u \in \mathbb{I}_i, v) \int_{\mathbb{I}_i} X(u) du \right)^2 dv}{\|X\|_2^2}.
\end{aligned} \tag{40}$$

We construct $\overline{X}_n(u \in \mathbb{I}_i) = n \int_{\mathbb{I}_i} X(u) du$, taking the mean of $X$ over each partition $\mathbb{I}_i$, where the length is $1/n$. Then the above equality becomes

$$\begin{aligned}
\frac{\left\| T_{\overline{W}_n} X \right\|_2^2}{\|X\|_2^2} &= \frac{\int_{[0,1]} \left( \sum_{i=1}^n \overline{W}_n(u \in \mathbb{I}_i, v) \overline{X}_n(u \in \mathbb{I}_i)/n \right)^2 dv}{\|X\|_2^2} \\
&= \frac{\left\| T_{\overline{W}_n} \overline{X}_n \right\|_2^2}{\|X\|_2^2}.
\end{aligned} \tag{41}$$

Comparing $\left\|\overline{X}_n\right\|_2^2$ and $\|X\|_2^2$, we have

$$
\begin{aligned}
\|X\|_2^2 &= \int_{[0,1]} X^2(u)du = \sum_{i=1}^n \int_{\mathbb{I}_i} X^2(u)du \\
&\geq \sum_{i=1}^n n \left( \int_{\mathbb{I}_i} X(u)1(u)du \right)^2 \\
&= \sum_{i=1}^n \frac{1}{n} \left( n \int_{\mathbb{I}_i} X(u)du \right)^2 \\
&= \left\|\overline{X}_n\right\|_2^2,
\end{aligned}
\tag{42}
$$

which is derived by the Cauchy-Schwarz inequality with $\int_{\mathbb{I}_i} 1^2(u)du = 1/n$. Therefore, for any $X \in L^2([0,1])$, it holds

$$
\frac{\left\|T_{\overline{\mathcal{W}}_n} X\right\|_2^2}{\|X\|_2^2} \leq \frac{\left\|T_{\overline{\mathcal{W}}_n} \overline{X}_n\right\|_2^2}{\left\|\overline{X}_n\right\|_2^2}.
\tag{43}
$$

For each $X \in L^2([0,1])$, we can derive a $\overline{X}_n \in L^2([0,1])$ from $X$ to make the above inequality hold. Then the operator norm becomes

$$
\left\|\overline{\mathcal{W}}_n\right\|_{2,2}^2 := \sup_{\forall \overline{X}_n \in L^2([0,1])} \frac{\left\|T_{\overline{\mathcal{W}}_n} \overline{X}_n\right\|_2^2}{\left\|\overline{X}_n\right\|_2^2},
\tag{44}
$$

although $\overline{X}_n$ is derived from $X$, obviously $\overline{X}_n$ can take all possible values within its defined space. Therefore, $\overline{X}_n$ can correspond to every $\boldsymbol{x}_n \in \mathbb{R}^n$.

$$
\begin{aligned}
\frac{\left\|T_{\overline{\mathcal{W}}_n} \overline{X}_n\right\|_2^2}{\left\|\overline{X}_n\right\|_2^2} &= \frac{\int_{[0,1]} \left( \sum_{i=1}^n \overline{\mathcal{W}}_n(u \in \mathbb{I}_i, v) \overline{X}_n(u \in \mathbb{I}_i)/n \right)^2 dv}{\left\|\overline{X}\right\|_2^2} \\
&= \frac{\sum_{j=1}^n \left( \sum_{i=1}^n \mathcal{S}_n(i,j) \boldsymbol{x}_n(i)/n \right)^2 /n}{\|\boldsymbol{x}\|_2^2/n} \\
&= \frac{1}{n^2} \frac{\|\mathcal{S}_n \boldsymbol{x}_n\|_2^2}{\|\boldsymbol{x}_n\|_2^2}.
\end{aligned}
\tag{45}
$$

As all $\boldsymbol{x}_n \in \mathbb{R}^n$ are considered in the above equality, we have

$$
\left\|\overline{\mathcal{W}}_n\right\|_{2,2}^2 := \sup_{\forall \boldsymbol{x}_n \in \mathbb{R}^n} \frac{1}{n^2} \frac{\|\mathcal{S}_n \boldsymbol{x}_n\|_2^2}{\|\boldsymbol{x}_n\|_2^2} = \frac{1}{n^2} \|\mathcal{S}_n\|_{2,2}^2.
\tag{46}
$$

$\square$

### D.1.2. CLASSICAL SIGNAL CONVERGENCE

**Lemma.** *(Convergence to Graphon Signal)*

*Consider the graphon signal $X_m$ generated from $X_{\mathbb{R}_+}$ that satisfies AS2, and the induced graphon signal $\overline{X}_{m,n}$ sampled from $X_m$, then the following holds*

$$
\Delta X_n \leq \frac{A_X t_m}{\sqrt{6n}\epsilon_u}
\tag{47}
$$

*with probability at least $1 - \epsilon_u$.*

*Proof.* The induced graphon signal $\overline{X}_n$ is defined on $n$ equal partitions of $[0,1]$, denoted by $\{\mathbb{I}_i | 1 \leq i \leq n\}$, i.e. $\mathbb{I}_i = [\frac{i-1}{n}, \frac{i}{n})$ for $1 \leq i \leq n-1$, $\mathbb{I}_n = [\frac{n-1}{n}, 1]$. Therefore, $\|\overline{X}_n - X\|_2$ can be divided into $n$ parts,

$$
\left\|\overline{X}_n - X\right\|_2^2 = \sum_{i=1}^n \int_{\mathbb{I}_i} (X(u_i) - X(u))^2 du.
\tag{48}
$$

By using Jensen's inequality and the $t_m A_X$-Lipschitz property, we have

$$
\begin{aligned}
\left\{ \mathbb{E}\left[\|\overline{X}_n - X\|_2\right] \right\}^2 &\leq \mathbb{E}\left[\|\overline{X}_n - X\|_2^2\right] \\
&= \mathbb{E}\left[\sum_{i=1}^n \int_{\mathbb{I}_i} (X(u_i) - X(u))^2 du\right] \\
&\leq A_X^2 t_m^2 \mathbb{E}\left[\sum_{i=1}^n \int_{\mathbb{I}_i} (u_i - u)^2 du\right].
\end{aligned}
\tag{49}
$$

For the expectation term of the above right side, calculate the integral of $u$,

$$
\begin{aligned}
\mathbb{E}\left[\sum_{i=1}^n \int_{\mathbb{I}_i} (u_i - u)^2 du\right] &= \mathbb{E}\left\{\sum_{i=1}^n \frac{i^3 - (i-1)^3}{3n^3} - \frac{2i-1}{n^2} u_i + \frac{1}{n} u_i^2\right\} \\
&= \frac{1}{3} - \frac{1}{n^2}\sum_{i=1}^n (2i-1)\mathbb{E}\left[u_i\right] + \frac{1}{n}\sum_{i=1}^n \mathbb{E}\left[u_i^2\right].
\end{aligned}
\tag{50}
$$

According to the order statistic theory over $n$ i.i.d. uniform distributions, the probability density of $u_i$ ($1 \leq i \leq n$) is $\text{Beta}(i, n-i+1)$ (Okoyo, 2016), that is

$$
\mathbb{E}[u_i] = \frac{i}{n+1}, \quad \mathbb{E}[u_i^2] = \frac{i^2 + i}{(n+1)(n+2)},
\tag{51}
$$

subtitute the above equations into (50), and calculate the sum,

$$
\mathbb{E}\left[\sum_{i=1}^n \int_{\mathbb{I}_i} (u_i - u)^2 du\right] = \frac{1}{6n}.
\tag{52}
$$

Combining (49) and (52), it becomes $\mathbb{E}\left[\|\overline{X}_n - X\|_2\right] \leq \frac{A_X t_m}{\sqrt{6n}}$. Using the Markov inequality, we have

$$
\mathrm{P}\left(\|\overline{X}_n - X\|_2 \geq \frac{A_X t_m}{\sqrt{6n}\epsilon_u}\right) \leq \epsilon_u.
\tag{53}
$$

$\square$

### D.1.3. CLASSICAL GRAPHON CONVERGENCE

**Lemma D.3.** *(Lei & Rinaldo, 2015) Considering the probability matrix $\mathbf{P}_n$ and the adjacency matrix $\mathbf{S}_n$ sampled from $\mathbf{P}_n$, for any $\nu > 0$, assume $n$ is large enough such that $n^{-\nu} \leq \epsilon_b$, there exists a constant $C_\nu$ such that*

$$
\|\mathbf{S}_n - \mathbf{P}_n\|_{2,2} \leq C_\nu \sqrt{n}
\tag{54}
$$

*with probability at least $1 - \epsilon_b$.*

**Lemma.** *(Convergence to Graphon)*

*Consider the sparse graphon $W_m$ generated from $W_{\mathbb{R}_+}$ that satisfies AS1, and the induced graphon $\overline{W}_{m,n}$ sampled from $W_m$. For any $\nu > 0$, assume $n$ is large enough such that $n^{-\nu} \leq \rho$, there exists a constant $C_\nu$ such that*

$$
\Delta T_{W_n} \leq \frac{C_\nu}{\sqrt{n}} + \frac{2A_W t_m}{\sqrt{6n}\epsilon_u}
\tag{55}
$$

*with probability at least $(1 - \epsilon_u)(1 - \rho)$.*

*Proof.* Based on the triangle inequality, $\Delta T_{W_n}$ can be divided into two parts:

$$
\Delta T_{W_n} \leq \left\|W - \overline{P}_n\right\|_{2,2} + \left\|\overline{P}_n - \overline{W}_n\right\|_{2,2}.
\tag{56}
$$

For $\left\|W - \overline{P}_n\right\|_{2,2}$, using Lemma 1 in (Ji et al., 2024) without stretching, we have

$$\left\|W - \overline{P}_n\right\|_{2,2} \leq \left\|W - \overline{P}_n\right\|_2. \tag{57}$$

Similar to the proof of Lemma D.1.2, we first divide $\left\|\overline{P}_n - W\right\|_2$ into $n^2$ parts, then employ the $t_m A_W$-Lipschitz property, it becomes

$$
\begin{aligned}
\left\|\overline{P}_n - W\right\|_2^2 &= \sum_{i=1}^n \sum_{j=1}^n \int\!\!\int_{\mathbb{I}_i \times \mathbb{I}_j} (W(u_i, u_j) - W(u, v))^2 du dv \\
&\leq A_W^2 t_m^2 \sum_{i=1}^n \sum_{j=1}^n \int\!\!\int_{\mathbb{I}_i \times \mathbb{I}_j} (|u_i - u| + |u_j - v|)^2 du dv \\
&\leq A_W^2 t_m^2 \sum_{i=1}^n \sum_{j=1}^n \int\!\!\int_{\mathbb{I}_i \times \mathbb{I}_j} 2(u_i - u)^2 + 2(u_j - v)^2 du dv \\
&= 4 A_W^2 t_m^2 \sum_{i=1}^n \int_{\mathbb{I}_i} (u_i - u)^2 du.
\end{aligned}
\tag{58}
$$

Using the same conclusion as in the proof of Lemma D.1.2, we have $\mathbb{E}\left[\left\|\overline{P}_n - W\right\|_{2,2}\right] \leq \frac{2 A_W t_m}{\sqrt{6n}}$. Then with probability at least $(1 - \epsilon_u)$, it holds

$$\left\|\overline{P}_n - W\right\|_{2,2} \leq \frac{2 A_W t_m}{\sqrt{6n \epsilon_u}}. \tag{59}$$

For $\left\|\overline{P}_n - \overline{W}_n\right\|_{2,2}$, combining Lemma D.2 and Lemma D.3, then for any $\nu > 0$, assume $n$ is large enough such that $n^{-\nu} \leq \rho$, there exists a constant $C_\nu$ such that

$$\left\|\overline{P}_n - \overline{W}_n\right\|_{2,2} \leq \frac{C_\nu}{\sqrt{n}}, \tag{60}$$

with probability at least $1 - \epsilon_b$. $\qquad\square$

## D.2. Stretched Form: Proof of Lemma 5.3

### D.2.1. BOUND OF STRETCHING COEFFICIENTS

**Lemma D.4.** *(Convergence of Stretching Coefficients)*

*For the $L^1$-norm of graphon $W_m$ and the induced graphon $\overline{W}_{m,n}$, it holds*

$$\left|\|W_m\|_1 - \|\overline{W}_{m,n}\|_1\right| \leq \Delta T_{W_n}. \tag{61}$$

*Proof.*

$$
\begin{aligned}
\left|\|\overline{W}_n\|_1 - \|W\|_1\right| &= \left|\int\!\!\int_{[0,1]^2} \overline{W}_n du dv - \int\!\!\int_{[0,1]^2} W du dv\right| \\
&= \left|\int_{[0,1]} \int_{[0,1]} (\overline{W}_n - W) \mathbf{1}(u) du \mathbf{1}(v) dv\right| \\
&\leq \left\|\int_{[0,1]} (\overline{W}_n - W) \mathbf{1}(u) du\right\|_2 \|\mathbf{1}(v)\|_2 \\
&\leq \left\|\overline{W}_n - W\right\|_{2,2} \|\mathbf{1}(u)\|_2 \\
&= \left\|\overline{W}_n - W\right\|_{2,2}.
\end{aligned}
\tag{62}
$$

The first inequality is derived from the Cauchy-Schwarz inequality, taking $\int_{[0,1]} (\overline{W}_n - W) \mathbf{1}(u) du$ as a function of $v$, and the second inequality is based on the definition of the operator norm. $\qquad\square$

That is $\left|s^2 - s_n^2\right| \leq \Delta T_{W_n}$ for stretching coefficients. When $n$ is sufficiently such that $\Delta T_{W_n} \leq s^2/2$, we have

$$\frac{1}{s_n^2} \leq \frac{2}{s^2}. \tag{63}$$

### D.2.2. STRETCHING EFFECTS ON GRAPHON AND SIGNAL

**Lemma D.5.** *Let* $(\mathcal{W}^{s_\forall}, X^{s_\forall})$ *be the stretched graphon and graphon signal of value-unbounded classical graphon* $\mathcal{W}$ *and classical graphon signal* $X$, *then it holds*

$$\|X^{s_\forall}\|_2 = \frac{1}{\sqrt{s_\forall}} \|X\|_2 \, ,$$

$$\|\mathcal{W}^{s_\forall}\|_{2,2} = \frac{1}{s_\forall} \|\mathcal{W}\|_{2,2} \tag{64}$$

*with non-specific stretching coefficient* $s_\forall \in (0, 1]$.

*Proof.* For stretched graphon signal $X^{s_\forall}$ with non-specific coefficient, consider its $L^2$-norm

$$\|X^{s_\forall}\|_2^2 = \int_{[0,+\infty)} \left(X^{s_\forall}(u)\right)^2 du = \int_{[0,1/s_\forall]} \left(X(s_\forall u)\right)^2 du$$

$$= \frac{1}{s_\forall} \int_{[0,1]} \left(X(u)\right)^2 du = \frac{1}{s_\forall} \|X\|_2^2 \, . \tag{65}$$

For stretched value-unbounded graphon $\mathcal{W}^{s_\forall}$, since it has all-zero regions where $u > 1/s_\forall$ or $v > 1/s_\forall$, for any $X_{\mathbb{R}_+} \in L^2(\mathbb{R}_+)$, we can find a corresponding stretched signal satisfying $X^{s_\forall}(u) = X(u)$ on $u \leq 1/s_\forall$ such that

$$\left\|T_{\mathcal{W}^{s_\forall}} X_{\mathbb{R}_+}\right\|_2^2 = \left\|T_{\mathcal{W}^{s_\forall}} X^{s_\forall}\right\|_2^2, \quad \left\|X_{\mathbb{R}_+}\right\|_2^2 \geq \left\|X^{s_\forall}\right\|_2^2. \tag{66}$$

Therefore, the operator norm can be transformed to

$$\|\mathcal{W}^{s_\forall}\|_{2,2}^2 = \sup_{\forall X_{\mathbb{R}_+} \in L^2(\mathbb{R}_+)} \frac{\left\|T_{\mathcal{W}^{s_\forall}} X_{\mathbb{R}_+}\right\|_2^2}{\left\|X_{\mathbb{R}_+}\right\|_2^2} = \sup_{\forall X^{s_\forall} \in L^2(\mathbb{R}_+)} \frac{\left\|T_{\mathcal{W}^{s_\forall}} X^{s_\forall}\right\|_2^2}{\left\|X^{s_\forall}\right\|_2^2}$$

$$= \sup_{\forall X \in L^2([0,1])} \frac{\left\|T_{\mathcal{W}^{s_\forall}} X^{s_\forall}\right\|_2^2}{\left\|X^{s_\forall}\right\|_2^2}. \tag{67}$$

Although $X^{s_\forall}$ is constructed from $X \in L^2([0,1])$, the function class $X^{s_\forall}$ lies in $L^2([0,+\infty))$ can still represent all square-integrable functions supported on the interval $[0, 1/s_\forall]$.

For the numerator, we consider the following:

$$\|T_{\mathcal{W}^{s_\forall}} X^{s_\forall}\|_2^2 = \int_{[0,1/s_\forall]} \left(\int_{[0,1/s_\forall]} \mathcal{W}^{s_\forall}(u,v) X^{s_\forall}(u) du\right)^2 dv$$

$$= \left(\frac{1}{s_\forall}\right)^3 \int_{[0,1]} \left(\int_{[0,1]} \mathcal{W}(u,v) X(u) du\right)^2 dv \tag{68}$$

$$= \left(\frac{1}{s_\forall}\right)^3 \|T_{\mathcal{W}} X\|_2^2 \, .$$

For the denominator, we have $\|X^{s_\forall}\|_2^2 = \frac{1}{s_\forall} \|X\|_2^2$. Substituting the above results into Eq.( 67), we obtain

$$\|\mathcal{W}^{s_\forall}\|_{2,2}^2 = \frac{1}{s_\forall^2} \sup_{\forall X \in L^2([0,1])} \frac{\|T_{\mathcal{W}} X\|_2^2}{\|X\|_2^2} = \frac{1}{s_\forall^2} \|\mathcal{W}\|_{2,2}^2 \, . \tag{69}$$

$\square$

**Lemma D.6.** *Consider different stretching coefficients $s_0, s_1 \in (0, 1]$, applied to the same classical graphon $W$ with signal $X$, respectively. Assume $\left|s_1^2 - s_0^2\right| \leq s_0^2/2$ ($s_0, s_1$ corresponding to $s_m, s_{m,n}$), then it holds*

$$\begin{aligned}
\left\|X^{s_0} - X^{s_1}\right\|_2 &= C_{X,1}\left|s_0^2 - s_1^2\right| + C_{X,2}\left|s_0^2 - s_1^2\right|^{1/2}, \\
\left\|W^{s_0} - W^{s_1}\right\|_{2,2} &= C_{W,1}\left|s_0^2 - s_1^2\right| + C_{W,2}\left|s_0^2 - s_1^2\right|^{1/2},
\end{aligned} \tag{70}$$

*where $C_{X,1}, C_{X,2}$ and $C_{W,1}, C_{W,2}$ are constants about graphon signal $X$ and graphon $W$, described in the following proof.*

*Proof.* Apply different non-specific stretching coefficients $s_0, s_1 \in (0, 1]$ to the same graphon signal $X_m$, which satisfies $A_X t_m$-Lipschitz. Using Lemma D.5, the difference between $X^{s_0}$ and $X^{s_1}$ becomes

$$\left\|X^{s_0} - X^{s_1}\right\|_2 = \frac{1}{\sqrt{s_0}}\left\|X - X^{s'}\right\|_2, \tag{71}$$

where $X^{s'}$ is stretched by the coefficient $s' = s_1/s_0$, that is $X^{s'}(u) = X(s'u)$ for $u \in [0, 1/s']$.

To compare the difference between $X \in L^2([0,1]^2)$ and $X^{s'}$ supported on $[0, 1/s']$, we need to discuss the effective integral area case-by-case:

$$\begin{cases} \textbf{Case 1:} & 1/s' \leq 1, \\ \textbf{Case 2:} & 1/s' \geq 1. \end{cases} \tag{72}$$

**Case 1:** When $1/s' \leq 1$, $\left\|X - X^{s'}\right\|_2$ expands to

$$\left\|X - X^{s'}\right\|_2^2 = \underbrace{\int_{[0,\frac{1}{s'}]}\left(X(u) - X^{s'}(u)\right)^2 du}_{\textbf{(i)}} + \underbrace{\int_{[\frac{1}{s'},1]}X^2(u)du}_{\textbf{(ii)}}. \tag{73}$$

The first part **(i)** represents the signal value difference, while the second part **(ii)** captures the area difference.

For the first part **(i)**, using the Lipschitz-property, we have

$$\begin{aligned}
\textbf{(i)} &= \int_{[0,\frac{1}{s'}]}\left(X(u) - X(s'u)\right)^2 du \\
&\leq \int_{[0,\frac{1}{s'}]}A_X^2 t_m^2\left(u - s'u\right)^2 du = \frac{A_X^2 t_m^2(1 - s')^2}{3s'^3}.
\end{aligned} \tag{74}$$

For the second part **(ii)**, the values of $X^2$ in $[\frac{1}{s'}, 1]$ is bounded by $\max(X^2)$. Therefore, we can directly calculate the area

$$\textbf{(ii)} = \int_{[\frac{1}{s'},1]}X^2(u)du \leq \max(X^2)\left(1 - \frac{1}{s'}\right). \tag{75}$$

Combining the results of **(i)** and **(ii)**, we have

$$\begin{aligned}
\left\|X - X^{s'}\right\|_2^2 &\leq \frac{A_X^2 t_m^2(1 - s')^2}{3s'^3} + \max(X^2)\left(1 - \frac{1}{s'}\right) \\
&\leq \frac{A_X^2 t_m^2(1 - s')^2}{3} + \max(X^2)\left|1 - \frac{1}{s'}\right|,
\end{aligned} \tag{76}$$

where the second inequality is derived by $1/s' \leq 1$.

**Case 2:** When $1/s' \geq 1$, $\left\|X - X^{s'}\right\|_2$ expands to

$$\left\|X - X^{s'}\right\|_2^2 = \underbrace{\int_{[0,1]}\left(X(u) - X^{s'}(u)\right)^2 du}_{\textbf{(i)}} + \underbrace{\int_{[1,\frac{1}{s'}]}X^{s'^2}(u)du}_{\textbf{(ii)}}. \tag{77}$$

Following a similar process to the above, we have

$$
\begin{aligned}
\textbf{(i)} &\le \int_{[0,1]} A_X^2 t_m^2 \left(u - s'u\right)^2 du = \frac{A_X^2 t_m^2 (1 - s')^2}{3}, \\
\textbf{(ii)} &\le \max(X^2) \left(\frac{1}{s'} - 1\right) = \max(X^2) \left|\frac{1}{s'} - 1\right|.
\end{aligned}
\tag{78}
$$

Combining the above two inequalities, we obtain

$$
\left\| X - X^{\mathfrak{s}'} \right\|_2^2 \le \frac{A_X^2 t_m^2 (1 - s')^2}{3} + \max(X^2) \left|\frac{1}{s'} - 1\right|,
\tag{79}
$$

The results of **case 1** and **case 2** share the same form, then we prove the final conclusion

$$
\begin{aligned}
\left\| X^{\mathfrak{s}_0} - X^{\mathfrak{s}_1} \right\|_2 &= \frac{1}{\sqrt{s_0}} \left\| X - X^{\mathfrak{s}'} \right\|_2 \\
&\le \frac{1}{\sqrt{s_0}} \left( \frac{A_X t_m}{\sqrt{3}} \left|1 - \frac{s_1}{s_0}\right| + \max(|X|) \left|1 - \frac{s_0}{s_1}\right|^{1/2} \right) \\
&= \frac{A_X t_m}{\sqrt{3} s_0^{3/2} (s_0 + s_1)} \left|s_0^2 - s_1^2\right| + \frac{\max(|X|)}{\sqrt{s_0 s_1 (s_0 + s_1)}} \left|s_0^2 - s_1^2\right|^{1/2}.
\end{aligned}
\tag{80}
$$

When $\left|s_1^2 - s_0^2\right| \le s_0^2/2$ for stretching coefficients, we have $\frac{1}{s_1^2} \le \frac{2}{s_0^2}$. The above inequality becomes

$$
\begin{aligned}
\left\| X^{\mathfrak{s}_0} - X^{\mathfrak{s}_1} \right\|_2 &\le \frac{A_X t_m}{\sqrt{3} s_0^{3/2} (s_0 + s_1)} \left|s_0^2 - s_1^2\right| + \frac{\max(|X|)}{\sqrt{s_0 s_1 (s_0 + s_1)}} \left|s_0^2 - s_1^2\right|^{1/2} \\
&\le C_{X,1} \left|s_0^2 - s_1^2\right| + C_{X,2} \left|s_0^2 - s_1^2\right|^{1/2},
\end{aligned}
\tag{81}
$$

where $C_{X,1} = \frac{A_X t_m}{\sqrt{3} s_0^{5/2} (1 + 1/\sqrt{2})}, C_{X,2} = \frac{\max(|X|)}{\sqrt{s_0^{5/2}/\sqrt{2}(1 + 1/\sqrt{2})}}$ are constans about $X$ and the stretching coefficient $s_0$.

Apply the two different non-specific stretching coefficients $s_0, s_1 \in (0, 1]$ to the same graphon $W$, which satisfies $A_W t_m$-Lipschitz, then consider the difference $\|W^{\mathfrak{s}_0} - W^{\mathfrak{s}_1}\|_{2,2}$, by Lemma D.5 and Lemma 1 in (Ji et al., 2024), it becomes

$$
\left\| W^{\mathfrak{s}_0} - W^{\mathfrak{s}_1} \right\|_{2,2} \le \frac{1}{s_0} \left\| W - W^{\mathfrak{s}'} \right\|_2,
\tag{82}
$$

where $W^{\mathfrak{s}'}$ is stretched based on the coefficient $s' = s_1/s_0$, that is $W^{\mathfrak{s}'}(u, v) = W(s'u, s'v)$ for $(u, v) \in [0, 1/s']^2$. Similarly, we analyze the difference under two distinct cases: $1/s' \le 1$ or $1/s' \ge 1$.

**Case 1:** When $1/s' \le 1$, $\left\| W - W^{\mathfrak{s}'} \right\|_2$ expands to

$$
\left\| W - W^{\mathfrak{s}'} \right\|_2^2 = \underbrace{\int \int_{[0, \frac{1}{s'}]^2} \left( W(u, v) - W^{\mathfrak{s}'}(u, v) \right)^2 dudv)}_{\textbf{(i)}}
$$
$$
+ \underbrace{\int \int_{[0,1]^2 - [0, \frac{1}{s'}]^2} W^2(u, v) dudv}_{\textbf{(ii)}}.
\tag{83}
$$

The value of $W$ in **(ii)** is bounded by 1. Following the same process used previously, we obtain

$$
\begin{aligned}
\textbf{(i)} &\le A_W^2 t_m^2 (1 - s')^2 \int \int_{[0, \frac{1}{s'}]^2} (u + v)^2 \, dudv = \frac{7 A_W^2 t_m^2 (1 - s')^2}{6 s'^4}. \\
\textbf{(ii)} &\le \int \int_{[0,1]^2 - [0, \frac{1}{s'}]^2} 1 dudv = 1 - \frac{1}{s'^2}.
\end{aligned}
\tag{84}
$$

Combining the above, we have the following:

$$
\begin{aligned}
\textbf{(i)} + \textbf{(ii)} &\leq \frac{7A_W^2 t_m^2 (1-s')^2}{6 s'^4} + \left(1 - \frac{1}{s'^2}\right) \\
&\leq \frac{7A_W^2 t_m^2 (1-s')^2}{6} + \left|1 - \frac{1}{s'^2}\right|,
\end{aligned}
\tag{85}
$$

where the second inequality is derived by $1/s' \leq 1$.

**Case 2:** When $1/s' \geq 1$, $\left\|W - W^{\mathfrak{s}'}\right\|_2$ expands to

$$
\left\|W - W^{\mathfrak{s}'}\right\|_2^2 = \underbrace{\int\int_{[0,1]^2} \left(W(u,v) - W^{\mathfrak{s}'}(u,v)\right)^2 dudv}_{\textbf{(i)}}
$$
$$
+ \underbrace{\int\int_{[0,\frac{1}{s'}]^2 - [0,1]^2} W^{\mathfrak{s}'2}(u,v) dudv}_{\textbf{(ii)}}.
\tag{86}
$$

Similarly, the value of $W^{\mathfrak{s}'}$ is bounded by $A_W(1-s')$. Then use the Lipschitz property to calculate **(i)**, and use the value bound to calculate **(ii)**:

$$
\textbf{(i)} \leq A_W^2 t_m^2 (1-s')^2 \int\int_{[0,1]^2} (u+v)^2 \, dudv = \frac{7A_W^2 t_m^2 (1-s')^2}{6}.
$$
$$
\textbf{(ii)} \leq \int\int_{[0,\frac{1}{s'}]^2 - [0,1]^2} 1 \, dudv = \frac{1}{s'^2} - 1.
\tag{87}
$$

Combining the above, we have the following.

$$
\textbf{(i)} + \textbf{(ii)} \leq \frac{7A_W^2 t_m^2 (1-s')^2}{6} + \left|\frac{1}{s'^2} - 1\right|.
\tag{88}
$$

Since results of **case 1** and **case 2** share the same form, we obtain

$$
\begin{aligned}
\|W^{\mathfrak{s}_0} - W^{\mathfrak{s}_1}\|_{2,2} &\leq \frac{1}{s_0} \left\|W - W^{\mathfrak{s}'}\right\|_2 \\
&\leq \frac{1}{s_0} \left(\sqrt{\frac{7A_W^2 t_m^2}{6}} |1 - s'| + \left|1 - \frac{1}{s'^2}\right|^{1/2}\right) \\
&\leq \frac{\sqrt{7} A_W t_m}{\sqrt{6} s_0^2 (s_0 + s_1)} \left|s_0^2 - s_1^2\right| + \frac{1}{s_0 \sqrt{s_1 (s_0 + s_1)}} \left|s_0^2 - s_1^2\right|^{1/2}.
\end{aligned}
\tag{89}
$$

When $\left|s_1^2 - s_0^2\right| \leq s_0^2/2$ for stretching coefficients, we have $\frac{1}{s_1^2} \leq \frac{2}{s_0^2}$. The above inequality becomes

$$
\|W^{\mathfrak{s}_0} - W^{\mathfrak{s}_1}\|_{2,2} \leq C_{W,1} \left|s_0^2 - s_1^2\right| + C_{W,2} \left|s_0^2 - s_1^2\right|^{1/2},
\tag{90}
$$

where $C_{W,1} = \frac{\sqrt{7} A_W t_m}{\sqrt{6} s_0^3 (1+1/\sqrt{2})}, C_{W,2} = \frac{1}{s_0^2 \sqrt{(1+1/\sqrt{2})/\sqrt{2}}}$ are constants about $W$ and the stretching coefficient $s_0$. $\qquad\square$

### D.2.3. STRETCHED GRAPH SEQUENCE CONVERGENCE (PROOF OF LEMMA 5.3)

For better distinction, we use $s$ and $s_n$ to denote different stretching coefficients of $W$ and $\overline{W}_n$. The differences of stretched graphons and stretched signals are derived by

$$
\left\|W^{\mathfrak{s}} - \overline{W}_n^{\mathfrak{s}_n}\right\|_{2,2} \leq \underbrace{\|W^{\mathfrak{s}} - W^{\mathfrak{s}_n}\|_{2,2}}_{\textbf{(a)}} + \underbrace{\left\|W^{\mathfrak{s}_n} - \overline{W}_n^{\mathfrak{s}_n}\right\|_{2,2}}_{\textbf{(b)}},
$$
$$
\left\|X^{\mathfrak{s}} - \overline{X}_n^{\mathfrak{s}}\right\|_2 \leq \underbrace{\|X^{\mathfrak{s}} - X^{\mathfrak{s}_n}\|_2}_{\textbf{(a)}} + \underbrace{\left\|X^{\mathfrak{s}_n} - \overline{X}_n^{\mathfrak{s}_n}\right\|_2}_{\textbf{(b)}},
\tag{91}
$$

where $W^{\mathfrak{s}_n}$ and $X^{\mathfrak{s}_n}$ are generalized stretched forms of $W$ and $X$ with coefficient $s_n$. Assume $n$ is sufficiently large such that $\left|s^2 - s_n^2\right| \le s^2/2$ in Lemma D.4, then applying the above Lemma D.5 and Lemma D.6, we obtain

$$\left\|W^{\mathfrak{s}} - \overline{W}_n^{\mathfrak{s}_n}\right\|_{2,2} \le \underbrace{C_{W,1}\left|s_0^2 - s_1^2\right| + C_{W,2}\left|s_0^2 - s_1^2\right|^{1/2}}_{(a)} + \underbrace{\frac{1}{s_n}\Delta T_{W_n}}_{(b)}$$

$$\le C_{W,1}\Delta T_{W_n} + C_{W,2}\Delta T_{W_n}^{1/2} + \frac{\sqrt{2}}{s}\Delta T_{W_n},$$

$$\left\|X^{\mathfrak{s}} - \overline{X}_n^{\mathfrak{s}}\right\|_2 \le \underbrace{C_{X,1}\left|s_0^2 - s_1^2\right| + C_{X,2}\left|s_0^2 - s_1^2\right|^{1/2}}_{(a)} + \underbrace{\frac{1}{\sqrt{s_n}}\Delta X_n}_{(b)} \tag{92}$$

$$\le C_{X,1}\Delta T_{W_n} + C_{X,2}\Delta T_{W_n}^{1/2} + \frac{2^{\frac{1}{4}}}{\sqrt{s}}\Delta X_n.$$

For simplicity, we leverage $C_{\mathfrak{s}}$ to represent the larger value of $\frac{\sqrt{2}}{s}$ and $\frac{2^{\frac{1}{4}}}{\sqrt{s}}$, that is

$$\left\|W^{\mathfrak{s}} - \overline{W}_n^{\mathfrak{s}_n}\right\|_{2,2} \le C_{\mathfrak{s}}\Delta T_{W_n} + C_{W,\mathfrak{s}}\Delta T_{W_n}^{1/2},$$

$$\left\|X^{\mathfrak{s}} - \overline{X}_n^{\mathfrak{s}}\right\|_2 \le C_{\mathfrak{s}}\Delta X_n + C_{X,\mathfrak{s}}\Delta T_{W_n}^{1/2}, \tag{93}$$

where $C_{\mathfrak{s}} := \max\{\frac{\sqrt{2}}{s}, \frac{2^{\frac{1}{4}}}{\sqrt{s}}\}$ is constant about $W$, $C_{W,\mathfrak{s}} = C_{W,1}\Delta T_{W_n}^{1/2} + C_{W,2}$ and $C_{X,\mathfrak{s}} = C_{X,1}\Delta T_{W_n}^{1/2} + C_{X,2}$ converge to constants $C_{W,2}$ and $C_{X,2}$ as $n$ increases.

# E. Convergence and Transferability of Filter and GWCN: Proof

Since we have already established the convergence of sparse graph models with respect to both the stretched graphon and the stretched signal, it follows that when the difference between filters (or GWCN outputs) is bounded by the differences of the underlying graphons and signals, the convergence of the filters and the GWCN naturally holds as well.

## E.1. Graph Convolutional Networks

Through the neighborhood structure defined by the graph topology, graph convolution allows each node to aggregate the features of its neighbors. As a signal processing framework on graphs, a graph convolutional operator (Segarra et al., 2017; Du et al., 2018; Gama et al., 2019) of $R$ order is defined as

$$h(\mathbf{S}_n)\boldsymbol{x}_n = \sum_{r=0}^{R} h_r \mathbf{S}_n^r \boldsymbol{x}_n, \tag{94}$$

where coefficients $h_r$ are weights of the $r$-th order convolution, aggregating features of each node's $r$-hop neighborhood.

Let $\Phi(\mathbf{S}_n, \boldsymbol{x}_n, \mathcal{H})$ denote GCN with learnable parameters $\mathcal{H}$, and 1-dimension input features $\boldsymbol{x}_n$ for simplicity. For an $L$-layers GCN, we denote the input/output feature dimension of each layer $l$ ($1 \le l \le L$) by $F^{l-1}/F^l$, with corresponding parameters $\mathcal{H}^l \in \mathbb{R}^{F^{l-1} \times F^l}$, which implies the relationship between input features and output features. For each layer $l$, the aggregation and propagation process can be expressed in channel-wise form (Ruiz et al., 2020; 2021b; 2023):

$$\boldsymbol{x}_{f_l} = \sigma\left(\sum_{f_{l-1}=1}^{F_{l-1}} h_{f_{l-1}, f_l}(\mathbf{S}_n)\boldsymbol{x}_{f_{l-1}}\right). \tag{95}$$

The activation function is denoted as $\sigma(\cdot)$. The learnable parameters $\mathcal{H}^l(f_{l-1}, f_l)$ are maintained in the weights of convolutional operators $h_{f_{l-1}, f_l}(\cdot) = \mathcal{H}^l(f_{l-1}, f_l)h(\cdot)$ in Eq.( 94), which is fixed during convergence and transferability.

### E.2. Difference Bound of Filter Outputs

**Lemma E.1.** *(Difference Bound of Filter Outputs)*

*Let $h(\cdot)$ denote the convolutional filter, applied to stretched graphons with signals $(W_1^{\mathfrak{s}}, X_1^{\mathfrak{s}})$ and $(W_1^{\mathfrak{s}}, X_1^{\mathfrak{s}})$, it holds*

$$\|h(W_1^{\mathfrak{s}})X_1^{\mathfrak{s}} - h(W_2^{\mathfrak{s}})X_2^{\mathfrak{s}}\|_2 \leq C_h \left(\Delta T_{W^{\mathfrak{s}}} \|X_1^{\mathfrak{s}}\|_2 + \Delta X^{\mathfrak{s}}\right), \tag{96}$$

*where $C_h$ is a constant related to the filter $h(\cdot)$ described in E.2.*

*Proof.* (**Proof of Lemma E.1.**)

Using the triangle inequality, the filter convergence error is divided into two parts:

$$\|h(W_1^{\mathfrak{s}})X_1^{\mathfrak{s}} - h(W_2^{\mathfrak{s}})X_2^{\mathfrak{s}}\|_2$$
$$\leq \underbrace{\|h(W_1^{\mathfrak{s}}) - h(W_2^{\mathfrak{s}})\|_{2,2} \|X_1^{\mathfrak{s}}\|_2}_{\textbf{(a)}} + \underbrace{\|h(W_2^{\mathfrak{s}})\|_{2,2} \Delta X^{\mathfrak{s}}}_{\textbf{(b)}}. \tag{97}$$

Parts **(a)** and **(b)** represent the operator difference and the signal difference, respectively.

Before proceeding to further analysis, we first consider the simplest case, namely the $1$-order shift operator $T_{W^{\mathfrak{s}}}$. For any stretched graphon $W^{\mathfrak{s}}$, its $L^1$-norm is equal to 1, i.e. $\|W^{\mathfrak{s}}\|_1 = 1$, then we have

$$\|W^{\mathfrak{s}}\|_{2,2}^2 \leq \int\int_{[0,+\infty)^2} (W^{\mathfrak{s}})^2 dudv$$
$$\leq \max(W^{\mathfrak{s}}) \int\int_{[0,+\infty)^2} W^{\mathfrak{s}} dudv \tag{98}$$
$$= \|W^{\mathfrak{s}}\|_1 = 1,$$

the first inequality is derived by Lemma 1 in (Ji et al., 2024). Recursively, we can derive the bound for the $r$-order operator, which is also valid for $W_2^{\mathfrak{s}}$, that is

$$\left\|T_{W^{\mathfrak{s}}}^{(r)}\right\|_{2,2} \leq 1. \tag{99}$$

For part **(a)**, we expand of the difference operator norm:

$$\|h(W_1^{\mathfrak{s}}) - h(W_2^{\mathfrak{s}})\|_{2,2} = \left\|\sum_{r=0}^{R} h_r \left(T_{W_1^{\mathfrak{s}}}^{(r)} - T_{W_2^{\mathfrak{s}}}^{(r)}\right)\right\|_{2,2}$$
$$\leq \sum_{r=0}^{R} h_r \left\|T_{W_1^{\mathfrak{s}}}^{(r)} - T_{W_2^{\mathfrak{s}}}^{(r)}\right\|_{2,2}, \tag{100}$$

the Cauchy-Schwarz inequality is used to derive the inequality. For the $r$-order shift operator, we have

$$\left\|T_{W_1^{\mathfrak{s}}}^{(r)} - T_{W_2^{\mathfrak{s}}}^{(r)}\right\|_{2,2}$$
$$= \left\|T_{W_1^{\mathfrak{s}}}\left(T_{W_1^{\mathfrak{s}}}^{(r-1)} - T_{W_2^{\mathfrak{s}}}^{(r-1)}\right) - \left(T_{W_1^{\mathfrak{s}}} - T_{W_2^{\mathfrak{s}}}\right)T_{W_2^{\mathfrak{s}}}^{(r-1)}\right\|_{2,2}$$
$$\leq \left\|T_{W_1^{\mathfrak{s}}}^{(r-1)} - T_{W_2^{\mathfrak{s}}}^{(r-1)}\right\|_{2,2} + \left\|T_{W_1^{\mathfrak{s}}} - T_{W_2^{\mathfrak{s}}}\right\|_{2,2} \tag{101}$$
$$\leq r \left\|T_{W_1^{\mathfrak{s}}} - T_{W_2^{\mathfrak{s}}}\right\|_{2,2}.$$

Substituting the above into the expression( 100), we obtain

$$\|h(W_1^{\mathfrak{s}}) - h(W_2^{\mathfrak{s}})\|_{2,2} \leq \left(\sum_{r=0}^{R} h_r r\right) \Delta T_{W^{\mathfrak{s}}}. \tag{102}$$

For part **(b)**, based on the properties of inequality( 99), we can derive:

$$\|h(W_2^{\mathfrak{s}})\|_{2,2} \le \sum_{r=0}^{R} h_r \left\| T_{W_2^{\mathfrak{s}}}^{(r)} \right\|_{2,2} \le \sum_{r=0}^{R} h_r. \tag{103}$$

By setting the constant $C_h := \max(\sum_{r=0}^{R} h_r r, \sum_{r=0}^{R} h_r) \, O(R^2)$, we complete the proof.

$\square$

### E.3. Difference Bound of GWCN Outputs

**Lemma E.2** (Difference of GWCN Outputs). *Consider the L-layer GWCN with learned parameters $\mathcal{H}$, where $F_0 = F_1 = 1$, applied to stretched graphons with signals $(W_1^{\mathfrak{s}}, X_1^{\mathfrak{s}})$ and $(W_2^{\mathfrak{s}}, X_2^{\mathfrak{s}})$, then*

$$\begin{aligned}
&\|\Phi(W_1^{\mathfrak{s}}, X_1^{\mathfrak{s}}, \mathcal{H}) - \Phi(W_2^{\mathfrak{s}}, X_2^{\mathfrak{s}}, \mathcal{H})\|_2 \\
&\le C(\mathcal{H}) \left(\|W_1^{\mathfrak{s}} - W_2^{\mathfrak{s}}\|_{2,2} \|X_1^{\mathfrak{s}}\|_2 + \|X_1^{\mathfrak{s}} - X_2^{\mathfrak{s}}\|_2\right),
\end{aligned} \tag{104}$$

*where $C(\mathcal{H})$ is a constant depending on $\mathcal{H}$ (proof in Appendix E.3).*

*Proof.* To distinguish between signals from different layers, we denote the output features of the $l$-th layer by $X_{f_L,1}^{\mathfrak{s}}$ and $X_{f_L,2}^{\mathfrak{s}}$, belonging to $\Phi(W_1^{\mathfrak{s}}, X_1^{\mathfrak{s}}, \mathcal{H})$ and $\Phi(W_2^{\mathfrak{s}}, X_2^{\mathfrak{s}}, \mathcal{H})$ respectively.

We analyze the GWCNs difference $\Delta\Phi^{\mathfrak{s}}$ from the last layer,

$$(\Delta\Phi^{\mathfrak{s}})^2 = \sum_{f_L=1}^{F_L} \left\| X_{f_L,1}^{\mathfrak{s}} - X_{f_L,2}^{\mathfrak{s}} \right\|_2^2. \tag{105}$$

According to the aggregation and propagation process of GWCNs, the output features can be expanded as

$$\begin{aligned}
X_{f_L,1}^{\mathfrak{s}} &= \sigma\left(\sum_{f_{L-1}}^{F_{L-1}} h_{f_{L-1},f_L}(W_1^{\mathfrak{s}}) X_{f_{L-1},1}^{\mathfrak{s}}\right), \\
X_{f_L,2}^{\mathfrak{s}} &= \sigma\left(\sum_{f_{L-1}}^{F_{L-1}} h_{f_{L-1},f_L}(W_2^{\mathfrak{s}}) X_{f_{L-1},2}^{\mathfrak{s}}\right).
\end{aligned} \tag{106}$$

Because the activation functions are normalized Lipschitz, that is $|\sigma(a) - \sigma(b)| \le |a - b|$, we derive the above

$$\begin{aligned}
\left\| X_{f_L,1}^{\mathfrak{s}} - X_{f_L,2}^{\mathfrak{s}} \right\|_2 &\le \left\| \sum_{f_{L-1}}^{F_{L-1}} h_{f_{L-1},f_L}(W_1^{\mathfrak{s}}) X_{f_{L-1},1}^{\mathfrak{s}} - \sum_{f_{L-1}}^{F_{L-1}} h_{f_{L-1},f_L}(W_2^{\mathfrak{s}}) X_{f_{L-1},2}^{\mathfrak{s}} \right\|_2 \\
&\le \sum_{f_{L-1}}^{F_{L-1}} \left\| h_{f_{L-1},f_L}(W_1^{\mathfrak{s}}) X_{f_{L-1},1}^{\mathfrak{s}} - h_{f_{L-1},f_L}(W_2^{\mathfrak{s}}) X_{f_{L-1},2}^{\mathfrak{s}} \right\|_2 \\
&\le \sum_{f_{L-1}}^{F_{L-1}} C_{\mathcal{H}} \left( \Delta T_{W^{\mathfrak{s}}} \left\| X_{f_{L-1},1}^{\mathfrak{s}} \right\|_2 + \left\| X_{f_{L-1},1}^{\mathfrak{s}} - X_{f_{L-1},2}^{\mathfrak{s}} \right\|_2 \right).
\end{aligned} \tag{107}$$

The second inequality is derived from the Cauchy-Schwarz inequality, and the third inequality is derived by Lemma E.1. The constant $C_{\mathcal{H}} \, O(R^2)$ is the maximum of all channel-wise filters $C_{h_{f_{l-1},f_l}}$, which in turn depends on $\mathcal{H}$.

Using the assumption about the activation function $\sigma(\cdot)$, we get $|\sigma(x)| = |\sigma(x) - 0| = |\sigma(x) - \sigma(0)| \le |x - 0| = |x|$, then

$\|X^{\mathfrak{s}}_{f_{L-1},1}\|_2$ can be derived by

$$
\begin{aligned}
\left\|X^{\mathfrak{s}}_{f_{L-1},1}\right\|_2 &\leq \left\|\sum_{f_{L-2}}^{F_{L-2}} h_{f_{L-2},f_{L-1}}(W^{\mathfrak{s}}_1)X^{\mathfrak{s}}_{f_{L-2},1}\right\|_2 \\
&\leq \sum_{f_{L-2}}^{F_{L-2}} C_{\mathcal{H}}\left\|X^{\mathfrak{s}}_{f_{L-2},1}\right\|_2 \\
&\leq \prod_{l=1}^{L-2} C_{\mathcal{H}}F_l \sum_{f_0}^{F_0}\left\|X^{\mathfrak{s}}_{f_0,1}\right\|_2 \\
&\overset{F_0=1}{=} \prod_{l=0}^{L-2} C_{\mathcal{H}}F_l\left\|X^{\mathfrak{s}}_1\right\|_2.
\end{aligned}
\tag{108}
$$

The second inequality is derived by considering the operator norm of filters in Lemma E.1, and the third inequality is obtained by calculating recursively. Substituting the inequality( 108) into the above result( 107), we have

$$
\left\|X^{\mathfrak{s}}_{f_L,1} - X^{\mathfrak{s}}_{f_L,2}\right\|_2 = \left(\prod_{l=0}^{L-1} C_{\mathcal{H}}F_l\right)\Delta T_{W^{\mathfrak{s}}}\left\|X^{\mathfrak{s}}_1\right\|_2 + C_{\mathcal{H}}\sum_{f_{L-1}}^{F_{L-1}}\left\|X^{\mathfrak{s}}_{f_{L-1},1} - X^{\mathfrak{s}}_{f_{L-1},2}\right\|_2,
\tag{109}
$$

expanded recursively, we have

$$
\left\|X^{\mathfrak{s}}_{f_L,1} - X^{\mathfrak{s}}_{f_L,2}\right\|_2 \leq L\left(\prod_{l=0}^{L-1} C_{\mathcal{H}}F_l\right)\Delta T_{W^{\mathfrak{s}}_1}\left\|X^{\mathfrak{s}}_1\right\|_2 + C^L_{\mathcal{H}}\sum_{f_0}^{F_0}\left\|X^{\mathfrak{s}}_{f_0,1} - X^{\mathfrak{s}}_{f_0,2}\right\|_2.
\tag{110}
$$

For simplicity, we have set $F_0 = 1$ and $F_L = 1$, then it holds

$$
\left\|\Phi(W^{\mathfrak{s}}_1,X^{\mathfrak{s}},\mathcal{H}) - \Phi(W^{\mathfrak{s}}_2,X^{\mathfrak{s}}_2,\mathcal{H})\right\|_2 \leq C(\mathcal{H})\left(\Delta T_{W^{\mathfrak{s}}}\left\|X^{\mathfrak{s}}_1\right\|_2 + \Delta X^{\mathfrak{s}}\right),
\tag{111}
$$

where $C(\mathcal{H}) := \max(C_1(\mathcal{H}),C_2(\mathcal{H})) \sim L(\prod_{l=0}^{L-1} F_l)O(R^{2L})$, $C_1(\mathcal{H}) = L\left(\prod_{l=0}^{L-1} C_{\mathcal{H}}F_l\right)$ and $C_2(\mathcal{H}) = C^L_{\mathcal{H}}$ are constants about the model parameters $\mathcal{H}$.

$\square$

### E.4. Convergence and Transferability of GWCNs

**Theorem E.3** (Convergence of GWCNs). *Consider an L-layer GWCN with learned parameters $\mathcal{H}$, denoted by $\Phi(W^{\mathfrak{s}},X^{\mathfrak{s}},\mathcal{H})$, where $F_0 = F_1 = 1$ for simplicity. Under assumptions from AS1 to AS3, and conditions on truncation length $t_m$ and graph size $n$ in Lemma 5.2, Lemma 5.3, and Lemma D.1, then for any $k' > 0$, it holds that*

$$
\Delta\Phi^{\mathfrak{s}}_{m,n} \leq C(\mathcal{H})\left(\frac{C_{\mathbb{R}_+}}{t^{k'}_m} + \frac{\left\|X_{\mathbb{R}_+}\right\|_{L^2([t_m,+\infty))}}{\left\|W_{\mathbb{R}_+}\right\|^{1/4}_1}\right) + C(\mathcal{H})\left(\frac{C_{m,1}}{n^{1/2}} + \frac{C_{m,2}}{n^{1/4}}\right).
\tag{112}
$$

*with probability at least $(1 - \epsilon_u)(1 - \epsilon_b)$. Here, $C(\mathcal{H})$ depends on $\mathcal{H}$, $C_{\mathbb{R}_+} = C_{k',W}\left\|X^{\mathfrak{s}}_{\mathbb{R}_+}\right\|_2 + C_{k',X}$, $C_{m,1} = C_{\mathfrak{s}}t_m(C_W\left\|X^{\mathfrak{s}}_m\right\|_2 + C_X)$, $C_{m,2} = (C_Wt_m)^{1/2}(C_{W,\mathfrak{s}}\left\|X^{\mathfrak{s}}_m\right\|_2 + C_{X,\mathfrak{s}})$.*

Based on Theorem E.3, we also prove Theorem 5.1 through a triangle inequality.

*Proof.* According to the sampling process of the sparse graph model, i.e. $W_{\mathbb{R}_+} \sim W_m \sim \overline{W}_{m,n}$, the same for $X_{\mathbb{R}_+} \sim X_m \sim \overline{X}_{m,n}$, the convergence error of graph sequences is divided into two parts:

$$
\textbf{Overall Error: } \Delta\Phi^{\mathfrak{s}}_{m,n} = \left\|\Phi(\overline{W}^{\mathfrak{s}}_{m,n},\overline{X}^{\mathfrak{s}}_{m,n},\mathcal{H}) - \Phi(W^{\mathfrak{s}}_{\mathbb{R}_+},X^{\mathfrak{s}}_{\mathbb{R}_+},\mathcal{H})\right\|_2,
$$

$$
\begin{cases}
\Delta\Phi^{\mathfrak{s}}_m = \left\|\Phi(W^{\mathfrak{s}}_m,X^{\mathfrak{s}}_m,\mathcal{H}) - \Phi(W^{\mathfrak{s}}_{\mathbb{R}_+},X^{\mathfrak{s}}_{\mathbb{R}_+},\mathcal{H})\right\|_2, \\
\Delta\Phi^{\mathfrak{s}}_n = \left\|\Phi(\overline{W}^{\mathfrak{s}}_{m,n},\overline{X}^{\mathfrak{s}}_{m,n},\mathcal{H}) - \Phi(W^{\mathfrak{s}}_m,X^{\mathfrak{s}}_m,\mathcal{H})\right\|_2.
\end{cases}
\tag{113}
$$

These two parts are divided by the triangle inequality, that is $\Delta\Phi_{m,n}^{\mathfrak{s}} \leq \Delta\Phi_m^{\mathfrak{s}} + \Delta\Phi_n^{\mathfrak{s}}$. Then based on Lemma E.2, we have

$$\Delta\Phi_{m,n}^{\mathfrak{s}} \leq C(\mathcal{H})\left(\Delta T_{W_m^{\mathfrak{s}}} \left\|X_{\mathbb{R}_+}^{\mathfrak{s}}\right\|_2 + \Delta X_m^{\mathfrak{s}}\right) + C(\mathcal{H})\left(\Delta T_{W_n^{\mathfrak{s}}} \|X_m^{\mathfrak{s}}\|_2 + \Delta X_n^{\mathfrak{s}}\right), \tag{114}$$

where $\Delta T_{W_m^{\mathfrak{s}}} = \left\|W_{\mathbb{R}_+}^{\mathfrak{s}} - W_m^{\mathfrak{s}}\right\|_{2,2}$ and $\Delta X_m^{\mathfrak{s}} = \left\|X_{\mathbb{R}_+}^{\mathfrak{s}} - X_m^{\mathfrak{s}}\right\|_2$. Let's review the above results of Lemma 5.2, Lemma D.1 and Lemma 5.3:

$$\begin{cases} \Delta T_{W_m^{\mathfrak{s}}} \leq \frac{C_{k',W}}{t_m^{k'}}, \\ \Delta X_m^{\mathfrak{s}} \leq \frac{C_{k',X}}{t_m^{k'}} + \frac{\|X_{\mathbb{R}_+}\|_{L^2([t_m,+\infty))}}{\sqrt{s_R}}. \end{cases} \tag{115}$$

$$\begin{cases} \Delta T_{W_n^{\mathfrak{s}}} \leq C_{\mathfrak{s}}\Delta T_{W_n} + C_{W,\mathfrak{s}}\Delta T_{W_n}^{1/2}, \\ \Delta X_n^{\mathfrak{s}} \leq C_{\mathfrak{s}}\Delta X_n + C_{X,\mathfrak{s}}\Delta T_{W_n}^{1/2}. \end{cases} \quad \begin{cases} \Delta T_{W_n} \leq \frac{C_W}{\sqrt{n}}t_m, \\ \Delta X_n \leq \frac{C_X}{\sqrt{n}}t_m. \end{cases}$$

Continuing the conditions and probabilities of these lemmas and substitute them into the above formula, we obtain

$$\begin{aligned} \Delta\Phi_{m,n}^{\mathfrak{s}} \leq\ & C(\mathcal{H})\left(\frac{C_{k',W}\left\|X_{\mathbb{R}_+}^{\mathfrak{s}}\right\|_2 + C_{k',X}}{t_m^{k'}} + \frac{\|X_{\mathbb{R}_+}\|_{L^2([t_m,+\infty))}}{\sqrt{s_R}}\right) \\ & + C(\mathcal{H})\left(\frac{C_{\mathfrak{s}}t_m(C_W\|X_m^{\mathfrak{s}}\|_2 + C_X)}{n^{1/2}} + \frac{(C_W t_m)^{1/2}(C_{W,\mathfrak{s}}\|X_m^{\mathfrak{s}}\|_2 + C_{W,\mathfrak{s}})}{n^{1/4}}\right), \end{aligned} \tag{116}$$

where the constant $C(\mathcal{H}) \sim L(\prod_{l=0}^{L-1} F_l)O(R^{2L})$ is related to the depth $L$, feature numbers $\{F_l\}$ and the convolutional order $R$ of the network framework. $\qquad\square$

