# OpenReview forum: "Size Transferability of Graph Convolutional Networks across Sparsity: A Generalized Graphon Perspective"
_ICML.cc/2026/Conference — ICML 2026 regular_

### Official Review · Reviewer_y8UB · 2026-03-03

**Soundness:** 3
**Presentation:** 3
**Significance:** 2
**Originality:** 3
**Overall Recommendation:** 4
**Confidence:** 3

**Summary:**

# Summary

This paper addresses a fundamental gap in the theoretical understanding of Graph Convolutional Networks (GCNs): proving size transferability across arbitrary sparsity regimes. The authors introduce Generalized Graphon Convolutional Networks (GWCNs) to establish a unified theoretical guarantee that transfer error vanishes as graph size increases and edge density decreases, regardless of fluctuating sparsity levels.

**The Research Gap**

* Existing GCN transferability analyses heavily rely on classical graphon limits, which inherently assume dense graphs where edge density remains constant at $\Theta(1)$.

* Real-world networks are sparse, meaning edge density diminishes as the graph scales.

* In sparse regimes, classical graphon limits trivially vanish to zero, rendering direct comparison meaningless.

* Furthermore, downsampling strategies used in size transfer create graph sequences with fluctuating, inconsistent sparsity levels.

* Prior theoretical frameworks either only apply to restricted sparsity (e.g., $\Omega(\log n / n)$), directly sample operators without explicit graph structures, or fail to guarantee a vanishing transfer error.

**Core Methodology: Generalized Graphons & Stretching**

* The authors model sparse random graphs using a double-sampling procedure from the generalized graphon framework.

* To prevent the limit from vanishing to zero, they employ a "stretching" operation.

* Stretching fixes the $L^{1}$-norm of the graphon to one by scaling it proportionally to its sparsity, which amplifies structural patterns and counteracts vanishing effects.

* GWCNs are introduced as the non-zero limit of GCNs under this stretched framework, providing a stable reference point for convergence across sequences with fluctuating densities.

**Key Theoretical and Empirical Contributions**

* **Theorem 5.1 (Transferability of GWCNs):** The authors derive an explicit upper bound on the transferability error between a source graph and a target graph across arbitrary sparsity levels.

* This error bound proves that topological information—specifically the interplay between expected edge density and graph size—governs the transfer error.

* The theory demonstrates that the expected edge density is the dominant factor in transferability, while graph size provides a secondary improvement.

* **Empirical Validation:** The authors test their claims on Cora, Pubmed, and Ogbn-Arxiv by downsampling subgraphs using three distinct sparsity schemes.

* The experiments corroborate the theoretical bounds, showing that transferability errors vanish consistently as edge densities decrease and graph sizes increase, converging across different sparsity trajectories.

**Compliance With Llm Reviewing Policy:**

Affirmed.

**Final Justification:**

Thank you for the additional clarification. This final response was helpful, and I appreciate the care with which you addressed the remaining points.

Regarding **[Q2]**, your latest explanation is more convincing than the earlier reply. In particular, it is helpful to understand that the Schwartz-space assumption was adopted mainly to guarantee $L^2$-convergence of $W_m$ and $X_m$ as $t_m \to \infty$, and that the broader mechanism is really governed by the decay of the $L^2$ tail integrals. I also appreciate the clarification that one could conceptually relax the assumption to a fixed-$k$ power-law form, or more generally to $L^2$-integrability. This improves my understanding of the modeling choice substantially. However, since these weaker assumptions are not part of the actual theorem proved in the current manuscript, I still view this concern as only partially resolved at the paper level.

Regarding **[Q3]**, I appreciate the explicit acknowledgement that global stretching in the theory does not perfectly mirror local degree normalization in practice. This is a useful clarification and improves the calibration of the paper’s claims. At the same time, it also confirms my earlier concern that the guarantee should be interpreted as being tied to the stretched theoretical framework, rather than as a direct guarantee for standard practical GCN normalization.

For **[Q4]**, your clarification about the role of degree-ordered sampling is helpful, and I agree that the connection to the generalized graphon truncation process makes the design choice more understandable. I also appreciate the explicit acknowledgement that caution is warranted when interpreting the smoothness of the convergence curves. This partially addresses my concern, though I still think the experimental discussion should make this dependence more explicit.

For **[Q5]** and **[Q6]**, I appreciate the commitment to integrating the additional comparative experiments and downstream performance metrics into the revised manuscript. These seem like meaningful additions. However, since they are not yet part of the reviewed version, I do not think they fully eliminate the empirical limitations I noted.

Overall, this additional response improves my understanding of the scope and intent of the work, especially on **[Q2]** and **[Q3]**. However, my remaining concerns still relate to the scope and interpretation of the central theoretical claims, and these are not issues that can be fully settled within a short rebuttal exchange. I therefore regard my concerns as partially resolved and keep my current assessment.

**Key Questions For Authors:**

- **\[Q1] Incremental Mathematical Contribution vs. Foundational Leap:** The formulation of the generalized graphon convolutional operator heavily relies on the signal processing framework recently established by Ji et al. (2024). Could the authors explicitly clarify which mathematical steps in Theorem 5.1 are fundamentally novel to the GWCN analysis, as opposed to being direct corollaries of applying a standard Lipschitz activation function to the existing linear filters from Ji et al. (2024)? What is the core technical hurdle overcome here that Ji et al. did not already solve?

- **\[Q2] The Contradiction in "Arbitrary Sparsity" (Assumption 1):** A core claim of this paper is the applicability to 'arbitrary sparsity of real-world networks'. However, Assumption 1 requires the generalized graphon to belong to the Schwartz space (rapidly decreasing). Given that many real-world sparse graphs (including citation networks like Cora and Pubmed used in Section 6) exhibit heavy-tailed, power-law degree distributions that decay polynomially, does AS1 practically exclude the very real-world networks this theory aims to model? How robust is Theorem 5.1 if AS1 is relaxed to accommodate power-law limits?

- **\[Q3] Theoretical Artifact vs. Practical Mechanism (The "Stretching" Normalization):** The paper achieves a vanishing transfer error by relying on a 'stretching' operation. In Section 6, this translates to scaling the GCN output by a factor of $\sqrt{2e\_{n}}/n$ rather than the classical $1/n$. Is the guaranteed transferability a property of the vanilla GCN architecture itself, or is it strictly dependent on modifying the GCN's aggregation normalization to be a function of the target graph's edge density? If a practitioner uses a standard mean-pooling or GCN normalization without this specific edge-density scaling, does the transfer error still vanish?

- **\[Q4] The "Degree Order" Sampling Artifact:** In Section 6, the expanding subgraphs are constructed by adding nodes 'in degree order'. This is a highly specific, deterministic sequence that likely induces heavy structural biases (e.g., forming a dense core before adding peripheral nodes). Real-world size transfer typically employs stochastic methods like random walks or neighborhood sampling. Does the empirical convergence hold if subgraphs are sampled uniformly or via random walks? How much of the smooth convergence in Figures 5-7 is an artifact of this degree-ordered construction rather than a general property of the networks?

- **\[Q5] Lack of Empirical Baselines:** While Table 1 provides a strong theoretical comparison of frameworks , the experiments exclusively plot the transferability difference of the proposed method. To truly demonstrate the practical superiority of GWCNs, could the authors provide an empirical comparison against a standard GCN or a recent baseline (such as the Dense-Sparse model by Ruiz et al., 2024)? It would greatly strengthen the paper to visually demonstrate the prior models failing (e.g., error converging to a constant) while the GWCN error successfully vanishes under Scheme I ($O(1/n)$).

- **\[Q6] Output Discrepancy vs. Downstream Performance:** The experimental evaluation measures transferability strictly via the average output error (embedding discrepancy) against the reference full-graph output. While this perfectly aligns with Theorem 5.1, it leaves a gap regarding practical utility. Does this vanishing $L^2$ error translate to preserved downstream task performance? Could the authors provide supplementary results showing how the actual node classification accuracy (or F1 score) on Cora/Pubmed changes as the graph size and sparsity scale?

- **\[Q7] Nomenclature Clarification (GWCN):** A minor point on terminology: the proposed model is named the "Generalized Graphon Convolutional Network," yet the acronym used is GWCN rather than GGCN.

**Limitations:**

The authors have partially addressed this requirement by including an Impact Statement at the end of the manuscript. They adequately discuss the societal impact, correctly identifying that as a foundational theoretical contribution, the work does not carry direct social risks, but rather indirectly supports the development of resource-efficient and robust algorithms. However, the authors have not adequately discussed the technical limitations of their proposed framework. It might be helpful if the authors can discuss the following points:

- **Acknowledge Theoretical Constraints:** The authors should discuss the restrictiveness of Assumption 1 (AS1), which requires the generalized graphon to belong to the Schwartz space (rapidly decreasing). They should be up front about how this mathematical constraint interacts with real-world scale-free networks that exhibit heavy-tailed, polynomial decay.

- **Acknowledge Empirical Constraints:** The authors should explicitly note the limitations of their experimental setup. Specifically, they should acknowledge that generating subgraphs by adding nodes in "degree order" is a deterministic heuristic that may not fully capture the structural nuances or noise of standard stochastic downsampling methods used in practice (e.g., random walks).

**Strengths And Weaknesses:**

# Strengths

- **\[S1] Strong Practical Motivation for a Theoretical Problem:** A major strength of this paper is how it bridges rigorous graph theory with a highly practical engineering bottleneck. Training Graph Convolutional Networks (GCNs) on massive real-world graphs incurs significant computational overhead. To mitigate this, practitioners frequently rely on "size transfer" heuristics—training a model on a smaller, downsampled subgraph and deploying it on a much larger target graph. However, because real-world networks are inherently sparse , the downsampling process creates subgraphs with fluctuating and inconsistent edge densities. Prior frameworks relying on classical graphons fail here because they implicitly assume dense graphs where edge density remains stable. By introducing the Generalized Graphon Convolutional Network (GWCN) , the authors provide a much-needed theoretical guarantee for this common heuristic across arbitrary sparsity levels. Ultimately, this work mathematically validates a widespread industry practice, facilitating more efficient and theoretically grounded model deployments in large-scale network environments.

- **\[S2] Theoretical Novelty and Breaking the Sparsity Barrier:** The theoretical contribution of this work is highly significant, successfully moving past the $\Omega(\log n / n)$ sparsity bottleneck that has constrained prior GCN transferability analyses. A well-known mathematical obstacle in sparse graph sequences is that their classical graphon limits trivially vanish to zero as $n \to \infty$, rendering standard convergence proofs useless. The authors provide an elegant and theoretically surprising solution by leveraging the generalized graphon framework alongside a dynamic "stretching" operation. By fixing the $L^1$-norm to one, stretching artificially preserves topological structures and allows for the construction of a meaningful, non-zero asymptotic limit: the Generalized Graphon Convolutional Network (GWCN).

- **\[S3] Explicit and Interpretable Error Bounds:** The formulation of Theorem 5.1 is particularly strong. Rather than settling for qualitative convergence, the authors derive an explicit upper bound on the transfer error that gracefully decomposes into size-dependent ($n, N$) and density-dependent ($t\_m, t\_M$) components. This provides deep analytical insight, rigorously proving that while expected edge density acts as the dominant factor governing transferability across arbitrary sparsity, graph size serves as a secondary regularizer.

- **\[S4] Robust Empirical Validation on Real-World, Extreme-Sparsity Regimes:** The experimental design is highly commendable for bridging rigorous theory with realistic network conditions. Rather than relying solely on synthetic graph generators that perfectly satisfy theoretical assumptions, the authors validate their claims on real-world datasets (Cora, Pubmed, and Ogbn-Arxiv) by approximating the expanding sequence through degree-ordered subgraphs. Most impressively, the authors explicitly test their model against three distinct sparsity trajectories, including the extreme $O(1/n)$ regime (Scheme I). Because prior frameworks fail to provide theoretical guarantees at this level of sparsity, demonstrating that the transfer error successfully vanishes under Scheme I provides compelling empirical proof of the GWCN framework's superiority. Furthermore, the experimental curves clearly illustrate the theoretical bounds established in Theorem 5.1, successfully disentangling the effects of edge density and graph size on the overall transfer error.


# Weaknesses

- See \[Key Questions For Authors]

---

> ### Author Rebuttal · Authors · 2026-03-31
>
> We sincerely thank reviewer y8UB for the rigorous evaluation.
>
> **( Figure link for Q4, Q5, Q6: [Anonymized Repository - Anonymous GitHub](https://anonymous.4open.science/r/icml-rebuttal-figs-2-3536/README.md) )**
>
> ---
>
> > **`[Q1]`: Incremental Mathematical Contribution vs. Foundational Leap**
>
> **R1**: We differ from Ji et al. 2024 in context and quantitative analysis. They explore generalized sparse models with qualitative convergence but no error bound, as it cannot be derived from their technique. We specifically tackle GCN transferability across sparsity by providing explicit **quantitative error bounds**. This is not merely a superposition of Lipschitz activation. Overcoming core technical hurdles, we derive novel bounds for discrepancies in stretching coefficients (Lemma D.4), operator norms and graph signals under random stretching (Lemma 5.2, 5.3, D.1, D.2, D.5).
>
> ---
>
> > **`[Q2]`: The Contradiction in Arbitrary Sparsity (AS1)**
>
> **R2**: Generalized graphon (AS1) and power-law degree distribution describe sparsity from distinct perspectives. The generalized graphon defines edge connection probability over a latent node space, where rapid decay means few nodes have high connection probabilities. Power-law degree distributions describe node degrees statistically. Structurally, the "tail" of a generalized graphon (nodes with low connection probability) corresponds to the "head" of a power-law distribution (majority of nodes with small degrees). Conversely, the power-law's tail corresponds to the graphon's head. Thus, they exist in different mathematical spaces and do not conflict in decaying rates.
>
> ---
>
> > **`[Q3] & L1`: Theoretical Artifact vs. Practical Mechanism (The Stretching Normalization)**
>
> **[R3]**: Both practical normalization (e.g., $\mathbf{D}^{-1}_n \mathbf{S}_n$) and our theoretical stretching factor fundamentally normalize based on node degrees. When analyzing size transferability, classical graphon use $1/n$ (Appendix Lemma D.2):
> $$
> \| \overline{ W } _ n \| _ {2,2}
> = \frac{ 1 }{  n } \| \mathbf{ S } _ n \| _ {2,2}.
> $$
> **This forces sparse graphs to a trivial zero limit.** To capture non-trivial topologies, stretching mechanism is introduced and modifies this scaling factor using the edge count (Appendix Lemma D.5):
> $$
> \| \overline{W} _ n^{\mathfrak{s}} \| _ {2,2}
> = \frac{1}{ \sqrt{2 e _ n }} \| \mathbf{S} _ n \| _ {2,2}
> = \frac{1}{ \sqrt{n \bar{d} _ n} } \| \mathbf{S} _ n \| _ {2,2},
> \nonumber
> $$
> where $\bar{d}_n$ is the average degree, $e_n$ is the total number of graph edges. Both practical local degree normalization and our theoretical average degree stretching share the same mechanism: incorporating edge density into the denominator to stabilize feature aggregation in sparse regimes. Stretching is therefore a theoretical abstraction of practical GCN normalization, not a mere artifact.
>
> ---
>
> > **`[Q4] & L2`: The Degree Order Sampling Artifact**
>
> **[R4]**: Degree-ordered sampling was employed for variable control, yielding non-overlapping sparsity curves (Schemes I-III) to isolate edge density impacts. Regarding empirical generalizability:
>
> 1. **Uniform sampling is included**: **Scheme III** represents unbiased subgraphs sampled uniformly. It has constant expected density, verifying that transfer error decreases as graph size increases.
>
> 2. **Supplementary Random Walk experiments**: We conduct random walk sampling on Cora with sizes 100-1200. Comparisons with Schemes I-III confirm that higher edge density consistently corresponds to higher transfer error. **(See link above).**
>
> Thus, practical sampling strategies corroborate our Theorem 5.1.
>
> ---
>
> > **`[Q5]`: Lack of Empirical Baselines:**
>
> **[R5]** In GCN size transferability, comparisons are typically theoretical due to incompatible experimental setups (details in **R4 to Reviewer QFCZ**). Notably, Ruiz et al. (2024) study spectral properties without analyzing GCN transfer error. In our comparative experiments with the closely related model, their convergence becomes increasingly difficult as sparsity intensifies, whereas our method consistently converges. **(See link above).**
>
> ---
>
> > **`[Q6]`: Output Discrepancy vs. Downstream Performance**
>
> **[R6]** As suggested, we supplement Cora experiments with cross-entropy, accuracy, and F1 score. Unlike continuous transfer error, classification metrics possess a margin of tolerance. If transfer error remains within decision boundaries, predictions are unchanged. Empirically, transfer error decreases smoothly while accuracy remains stable, with reduced variance. This verifies that the "bounded error tending to zero" (Theorem 5.1) translates directly to stable downstream performance. **(See link above).**
>
> ---
>
> > **`[Q7]`: Nomenclature Clarification (GWCN)**
>
> **[R7]**: We use $W$ for graphons to distinguish them from $G$ (graphs), adhering to established conventions in this research field.
>
> ---
>
> We sincerely thank reviewer y8UB again for the time, expertise, and constructive suggestion.

---

> > ### Author Rebuttal · Reviewer_y8UB · 2026-04-01
> >
> > Thank you for the detailed rebuttal. It clarified several aspects of the paper, and in particular it helped me better understand the intended novelty relative to Ji et al. (2024). Regarding [Q1], I now understand the main claimed novelty to be the explicit quantitative transfer bound for GCN transferability across sparsity, together with the new control of stretching/operator/signal discrepancies, rather than the generalized graphon operator itself. This substantially clarifies my original concern.
> >
> > That said, my main reservations are only partially resolved.
> >
> > For [Q2], the manuscript still relies on a strong assumption that the generalized graphon belongs to the Schwartz space. Your response clarifies that generalized-graphon decay and power-law degree distributions live in different mathematical spaces, which is helpful conceptually, but it does not fully resolve my underlying concern about how restrictive AS1 is for realistic sparse-network models or how robust the theorem is if AS1 is weakened. Relatedly, I also find reviewer b2Wc’s point relevant here: the transfer bound depends on an additional signal-tail condition, which further narrows the effective applicability of the result.
> >
> > For [Q3], I appreciate the explanation that stretching should be viewed as a theoretical abstraction of practical degree-based normalization. However, I still do not think the rebuttal fully answers the practical counterfactual in my question, namely whether the same vanishing-transfer conclusion should be expected for a practitioner using a standard GCN normalization rather than the stretched regime analyzed in the paper. My current reading remains that the guarantee is fundamentally tied to the stretched formulation.
> >
> > For [Q4], your reply is partially helpful. I agree that the current experimental setup is not purely deterministic, since the sampled graphs are still formed by random node selection. However, the expanding stage sequence is still constructed in degree order, so I still think some caution is warranted in interpreting the smooth empirical convergence as fully sampling-agnostic. The additional random-walk discussion is useful, but this point is only partially closed for me.
> >
> > For [Q5] and [Q6], the rebuttal improves the paper’s empirical case, but only partially. I understand the argument that cross-framework empirical comparisons are difficult because prior works use incompatible graph models, and the added downstream metrics are relevant to my concern. Still, these additions mainly appear in the rebuttal rather than the paper itself, so they reduce but do not eliminate my reservations about the empirical support.
> >
> > The naming issue in [Q7] is satisfactorily clarified.
> >
> > Overall, the rebuttal strengthens my understanding of the contribution and addresses some of my questions, but my remaining concerns are about the scope and interpretation of the central theoretical claims, rather than details that can be fully settled in a short rebuttal. For that reason, I view the rebuttal outcome as only partially resolving my concerns, and I am keeping my current assessment.

---

> > > ### Author Response · Authors · 2026-04-04
> > >
> > > Dear Reviewer y8UB,
> > >
> > > We sincerely thank you for your time, expertise, and detailed attention to our manuscript and rebuttal. Your feedback has provided us with valuable inspiration and deeper insights into our work.
> > >
> > > We are pleased that our responses helped clarify your questions regarding **[Q1]** and **[Q7]**, and partially addressed your concerns for **[Q3]–[Q6]**. We would like to take this opportunity to provide *a few additional clarifications.*
> > >
> > > ---
> > >
> > > Regarding **[Q2]**, **your insights have offered us new perspectives.**
> > >
> > > Our motivation for employing the Schwartz space assumption for the generalized graphon and the tail integral assumption for the generalized graphon signal, is to guarantee the **convergence in the $L^2$-norm** for $W_m$ and $X_m$ (defined on the $[0, t_m]$ interval) as $t_m \to \infty$.
> > >
> > > In our theoretical proofs, we utilize the polynomial upper bound property of the Schwartz function:
> > > $$
> > > \forall k > 0, W _ { \mathbb{R} _ + } \leq \frac{ C _ k }{ (u + 1)^k (v + 1)^k }
> > > \quad \to \quad
> > > \forall k > 0, W_{m} \text{ converge to } W _ {\mathbb{R} _ +}.
> > > $$
> > > *If we were to relax the functional requirements, the Schwartz assumption could be generalized to a power-law assumption:*
> > > $$
> > > \text{For a fixed } k > 0, W _ {\mathbb{R} _ +} = \frac{ C _ k }{ (u + 1)^k (v + 1)^k }
> > > \quad \to \quad
> > > W _ {m} \text{ converge to } W _ {\mathbb{R} _ +}.
> > > $$
> > > In this case, the decay rate of the bound would depend on that specific $k$ rather than $\forall k > 0$. **In the broadest sense, this convergence only requires $L^2$-integrability.** The bound in Theorem 5.1 would then be governed by the decay rate of the $L^2$ tail integrals of $W$ and $X$:
> > > $$
> > > \Delta \Phi
> > > \sim
> > > L_W^2(t_m) + L_X^2(t_m) +O(n^{-1/4}),
> > > $$
> > > where $L_W^2(t_m)$ and $L_X^2(t_m)$ represent the $L^2$ tail integrals over $\mathbb{R}_+^2 \setminus [0, t_m]^2$ and $[t_m, +\infty)$, respectively.
> > >
> > > ---
> > >
> > > Regarding **[Q3]**, we appreciate your insightful feedback, global stretching in our theory **does not perfectly mirror** local degree normalization in practice.
> > >
> > > **We adopted the stretching framework to overcome the limitations of prior works in handling extreme sparsity**, allowing us to provide a unified convergence guarantee across "arbitrary sparsity". We will clarify the connection and distinction between this theoretical abstraction and practical deployment in the updated paper.
> > >
> > > ---
> > >
> > > Regarding **[Q4]**, we thank you for the reminder and **we agree with your view that caution is needed when interpreting the smoothness of the empirical convergence curves.**
> > >
> > > The degree ordered sampling does introduce structural smoothness. **This design is chosen to correspond to the mathematical process in generalized graphon model, where the truncation interval expands from a "dense core" to a "sparse tail", on real finite graphs.** Meanwhile, as discussed before, *our uniform sampling (Scheme III) and supplementary random walk experiments are independent of the degree ordered setting and were conducted directly on the original full graph*. Their results verify the conclusion that "density affects error". Following your suggestion, we will add a discussion in the revised experimental section to clarify the specific impact of different sampling strategies.
> > >
> > > ---
> > >
> > > Regarding **[Q5]** and **[Q6]**, we thank you for the suggestions. We agree that it is meaningful to **add these empirical results from the rebuttal into the main text**.
> > >
> > > The new performance metrics in **[Q6]** are built on our existing experiments , and the comparative experiments in **[Q5]** were conducted on straightforward synthetic datasets. *These additions can be integrated to provide more robust support for our claims while maintaining the original focus of the manuscript.* We commit to adding them in the updated version.
> > >
> > > ---
> > >
> > > We sincerely thank you for your professionalism and deep engagement throughout the review process. Your feedback is insightful and these discussions are meaningful for improving the rigor of the paper.
> > >
> > > Best regards,
> > >
> > > The Authors of Submission 15419

---

### Official Review · Reviewer_QFCZ · 2026-03-07

**Soundness:** 3
**Presentation:** 2
**Significance:** 3
**Originality:** 3
**Overall Recommendation:** 4
**Confidence:** 3

**Summary:**

This paper proposes the Generalized Graphon Convolutional Network (GWCN) based on generalized graphon theory to study the size transferability of Graph Convolutional Networks (GCNs) across varying sparsity levels. By introducing a stretching operation, GWCN constructs a non-trivial limit that preserves graph topology in sparse settings. The authors further provide both theoretical analysis and empirical validation of a transfer error bound, characterizing its dependence on graph size and edge density.

**Compliance With Llm Reviewing Policy:**

Affirmed.

**Final Justification:**

The authors have addressed most of my concerns. I am willing to increase my score.

**Key Questions For Authors:**

Please refer to the Weaknesses.

**Limitations:**

Please refer to the Weaknesses.

**Strengths And Weaknesses:**

Strengths:

1. This paper introduces the Generalized Graphon framework to handle graph sequences with arbitrary sparsity, and proposes the Generalized Graphon Convolutional Network (GWCN) as the non-zero limit of GCNs, overcoming the previous restriction of analyses being limited to dense or relatively sparse graphs.

2. In the theoretical part, the paper develops a comprehensive framework for convergence and transferability analysis, including the generation of sparse graph sequences, the introduction of generalized graphons, the definition of GWCN, and the derivation of explicit bounds on transfer errors.

Weaknesses:

1. The paper suffers from limited readability and insufficient explanation of symbols. Some notations and terms are not clearly defined in the text, making it difficult to follow.

2. Experimental details are incomplete, limiting reproducibility. The paper does not provide full experimental settings, hyperparameter choices, or code. Providing code and a detailed parameter table would enhance the reliability of the method and facilitate follow-up research.

3. Experimental validation is limited. The experiments are conducted only on 2–3 layer GCNs with hidden sizes of 32 and 64, without testing other network depths, hidden dimensions, activation functions, or alternative GNN architectures such as GraphSAGE or GAT. This restricts the assessment of the method’s generalization and the verification of theoretical claims.

4. There is a lack of direct empirical comparison with existing methods. Although multiple theoretical frameworks are discussed in the introduction and Table 1, the experiments do not compare GWCN performance against existing approaches, leaving the claimed advantages unverified empirically.

---

> ### Author Rebuttal · Authors · 2026-03-31
>
> We sincerely thank reviewer QFCZ for the constructive feedback, regarding readability, experimental reproducibility, and empirical validation.
>
> ---
>
> > **`W1`:  The readability and explanation**
>
> **R1**: Thank you for the suggestion. We will systematically reorganize the manuscript content, verify notation and symbol definitions, and provide additional explanations regarding their physical significance to improve overall readability.
>
> ---
>
> > **`W2`: Experimental reproducibility**
>
> **R2**: Focuses on fundamental GCN theory, we intentionally utilize straightforward parameter settings for training and testing. This demonstrates that the vanishing transfer error is a general theoretical property rather than an artifact of specific task tuning (notably, some paper in this field even uses **untrained parameters**, e.g., [5]).
>
> Due to the ICML regulations stating that "**links may only be used for figures**," and AC confirmed that providing an anonymous code link is not permitted. Regarding the experimental setup, models are trained on randomly sampled subgraph of medium size, using a 50%/50% train/test split. The trained GCN is then applied to sampled subgraphs of varying sizes to measure transfer errors. For hyperparameters, we employ the Adam optimizer with learning rate 0.08 and weight decay 5e-4 for 100 or 200 training epochs.
>
> We commit to providing code in the latter permitted stage and will incorporate comprehensive experimental details into the revised manuscript.
>
> ---
>
> > **`W3`: Experimental validation**
>
> **R3**: In GCN size transferability research  [1]-[10], employing 2/3 layers with hidden dimensions of 32/64 is a general setting to verify theoretical conclusions *(shown in table below)*. Thus, we adhere to these settings. Furthermore, our Ogbn-Arxiv experiments in the Appendix also use **larger hidden dimensions of 128 and 256**.
>
> Our theoretical conclusions are established specifically for GCNs, which operate on fixed graph topologies. In contrast, GraphSAGE's neighbor sampling and GAT's dynamic attention mechanisms reconstruct topologies. Such mechanisms fall outside the scope of this work. As noted in R2 to Reviewer exv6, extending this framework to other GNNs is a primary objective for our future research.
>
> ---
>
> > **`W4`: Empirical comparison with existing methods**
>
> **R4:** In this field [1]-[10], the primary contribution is theoretical analysis based on graph model convergence, with experiments designed within those models to verify the theory. **Different methodologies utilize distinct graph models, convergence approaches, and theoretical frameworks.** Consequently, comparisons in related research are typically conducted only from a theoretical perspective; experimental comparisons are generally avoided due to the difficulty of constructing compatible experimental schemes [1]-[10]  (shown in **table** below).
>
> **The differences in graph models** are as follows: **[1] [2]** focus on dense graphs and are inapplicable to sparse scenarios. **[3]** does not perform experiments. **[4] [9]** directly discretize graph operators without constructing explicit graph sequences. **[5]** utilizes a graphon model with restricted value ranges. **[6]** employs the same graph model as [5] but does not focus on sparsity. **[7] [8]** construct weighted graphs, whereas ours are unweighted. **[10]** primarily focuses on spectral properties and does not analyze transfer error.
>
> Therefore, we chose to conduct comparison with **[5]** and fully inherited their settings: Additionally, we introduce sparser $\Theta(1/n)$ and $\Theta(\log n / n)$ scenarios. Results demonstrate that as sparsity intensifies, their convergence becomes increasingly difficult, while our model consistently converges.
>
> **(Figures are here: [Anonymized Repository - Anonymous GitHub](https://anonymous.4open.science/r/icml-rebuttal-figs-7547)).**
>
> ---
>
> | Models in Works | Datasets | GCN Architectures | Empirical Comparison across Theoretical Frameworks |
> | :--- | :--- | :--- | :--- |
> | Classical Graphon [1] [2] | Cora, MovieLens, Decentralized Flocking | 1-2 layers, 8/32/64 hidden | None |
> | Unbounded Graphon [3] | --- | --- | None (Pure Theory) |
> | Continuous Space [4] | Bunny mesh, MNIST, Citeseer, Cora | 3 layers, 32/64 hidden | None |
> | Bounded Graphon [5] [6] | Synthetic graphs (e.g., SBMs) | Untrained (omitted) | None |
> | Geometric Graph [7] [8] | Flocking simulation, ModelNet40 | 1-3 layers, 32-128 hidden | None |
> | Operator and Graphop [9] | Synthetic graphs (e.g., Polymer graphs)| 2 layers (hidden omitted) | None |
> | Dense-Sparse Model [10] | Synthetic graphs (e.g., DSGM), Chameleon| (omitted) | None |
> | **Generalized Graphon (Ours)**| **Cora, Pubmed, Ogbn-Arxiv** | **2/3 layers, 32/64/128/256 hidden**| **Bounded Graphon** [5] |
>
> ---
>
> **Note: References** are in our response to Reviewer exv6, due to the length limit.
>
> ---
>
> We sincerely thank you again for the valuable time and constructive feedback provided during this review.

---

> > ### Author Rebuttal · Reviewer_QFCZ · 2026-04-01
> >
> > The authors have addressed most of my concerns. I am willing to increase my score.

---

> > > ### Author Response · Authors · 2026-04-02
> > >
> > > Dear Reviewer QFCZ,
> > >
> > > Thank you for your generous acknowledgment of our response.
> > >
> > > We are glad that our responses regarding readability, experimental reproducibility, and empirical validation have successfully addressed your concerns. We will incorporate these clarifications, along with our code and experimental details, into our manuscript.
> > >
> > > Best regards,
> > >
> > > The Authors of Submission 15419

---

### Official Review · Reviewer_exv6 · 2026-03-07

**Soundness:** 3
**Presentation:** 3
**Significance:** 3
**Originality:** 3
**Overall Recommendation:** 4
**Confidence:** 3

**Summary:**

This paper studies the transferability of GCNs across varying sparsity levels. Subgraphs are randomly generated from the original graph, resulting in multiple sparsity. By introducing GWCNs as the non-zero limit of GCNs, this paper establishs a transfer error bound over sparse graphs, which is influenced by expected edge density and graph size, factors that jointly determine the multiple sparsity.

**Compliance With Llm Reviewing Policy:**

Affirmed.

**Final Justification:**

Although the authors did not directly address all the specific concerns raised in my review, particularly regarding the formal convergence proof for the entire size transfer learning process and the extension to more modern GNN architectures, I maintain my original score based on the paper's strengths and the overall quality assessment.

**Key Questions For Authors:**

see Weaknesses

**Limitations:**

yes

**Strengths And Weaknesses:**

Strengths

1. The paper directly confronts a well-known limitation of classical graphon-based analyses, their inapplicability to arbitrarily sparse graphs. This is a fundamental problem for the theoretical understanding of GCNs on real-world data, making the work highly relevant.

2. The explicit transfer error bound is a key strength, as it clearly breaks down the error into parts driven by graph size and edge density, offering practical insights into how these factors affect transfer performance.


Weaknesses

1. While the paper establishes convergence for the underlying graph sequences and provides a bound on the output discrepancy of GWCNs, it does not offer a formal convergence proof for the entire size transfer learning process. Specifically, there is no analysis showing that the training dynamics on the source subgraph lead to parameters that are guaranteed to perform well on the target graph within the derived bound.

2. The analysis is focused specifically on standard spectral-domain GCNs. It remains an open question whether the same framework and guarantees can be extended to more modern and powerful GNN architectures (e.g., those with attention mechanisms like GAT).

---

> ### Author Rebuttal · Authors · 2026-03-31
>
> We thank reviewer exv6 for the insightful feedback and the opportunity to clarify the scope and future directions of our work. Your comments regarding the training process and the applicability to diverse architectures are highly valuable.
>
> ---
>
>
> > `W1`: **The analysis about training process.**
>
> **R1**: Thank you for the suggestion regarding the training perspective.
>
> Within the research field of GCN size transferability [1]-[9], GCN parameters are considered already trained and fixed, then GCN output discrepancy is analyzed when it's applied to graphs of different sizes. Therefore, our work primarily focuses on **inference transferability** instead of training dynamics. From a theoretical perspective, for any fixed GCN parameters, the resulting discrepancy across different graphs should fall within the derived transfer bound.
>
> Analyzing GCN training dynamics, which involves the complex evolution of parameters during the learning process, represents a significant and valuable direction that we look forward to exploring in future research.
>
> ---
>
> > `W2`: **Extension to GNNs with attention mechanisms like GAT**
>
> **R2**: Thank you for this forward-looking question. While our current GCN conclusions are not yet directly applicable to GNNs with attention mechanisms like GAT, our theoretical framework provides a necessary baseline for such extensions.
>
> The multi-layer convolutions of GCNs rely on a fixed graph topology for neighborhood aggregation. In constrast, models like GAT utilize attention mechanisms to dynamically adjust edge weights and reconstruct the topology based on node features. Despite these differences, both architectures essentially rely on neighborhood aggregation to achieve information propagation.
>
> In this paper, we have derived the GCN transfer error over sparse graph sequences, where the topology within the GCN remains constant. This provides a fundamental baseline for future research to investigate how transferability properties evolve when the topology within the GNN becomes dynamic.
>
> ---
>
> **References:**
>
> [1] Ruiz et al. (2020). Graphon neural networks and the transferability of graph neural networks.
>
> [2] Ruiz et al. (2023). Transferability properties of graph neural networks.
>
> [3] Maskey et al. (2023). Transferability of graph neural networks: An extended graphon approach.
>
> [4] Levie et al. (2021). Transferability of spectral graph convolutional neural networks.
>
> [5] Keriven et al. (2020). Convergence and stability of graph convolutional networks on large random graphs
>
> [6] Keriven et al. (2021). On the Universality of Graph Neural Networks on Large Random Graphs
>
> [7] Wang et al. (2022). Convergence of graph neural networks on relatively sparse graphs.
>
> [8] Wang et al. (2023). Geometric graph filters and neural networks: Limit properties and discriminability trade-offs.
>
> [9] Le & Jegelka, (2023). Limits, approximation and size transferability for gnns on sparse graphs via graphops.
>
> [10] Ruiz et al. (2024). A spectral analysis of graph neural networks on dense and sparse graphs.
>
> ---
>
> We thank reviewer exv6 again for the time and the helpful comments.

---

> > ### Author Rebuttal · Reviewer_exv6 · 2026-04-02
> >
> > I thank the authors for the detailed rebuttal. I maintain my score.

---

> > > ### Author Response · Authors · 2026-04-02
> > >
> > > Dear Reviewer exv6,
> > >
> > > Thank you for your constructive evaluation and for confirming that your concerns are fully resolved. We are delighted that our discussion on inference transferability and attention-based GNNs successfully addressed your questions.
> > >
> > > We deeply value your forward-looking suggestions, which provide inspiration for our future work.
> > >
> > > Best regards,
> > >
> > > The Authors of Submission 15419

---

### Official Review · Reviewer_b2Wc · 2026-03-11

**Soundness:** 3
**Presentation:** 3
**Significance:** 3
**Originality:** 3
**Overall Recommendation:** 4
**Confidence:** 5

**Summary:**

This paper studies GCN transferability across arbitrary sparsity regimes. It introduces the Generalized Graphon Convolutional Network (GWCN) as the non-trivial limit of GCNs under the generalized graphon framework, where sparse graph sequences are constructed via truncation and rescaling of a generalized graphon. The main result is an explicit transfer error bound that decomposes into a density dependent term (governed by truncation length $t_m$) and a size-dependent term (governed by $n$), vanishing as both grow. Experiments on Cora, PubMed, and Ogbn-Arxiv validate the theoretical findings.

**Compliance With Llm Reviewing Policy:**

Affirmed.

**Final Justification:**

The authors have only partially addressed my concerns related to the tightness of their transferability bound, but I am willing to retain my previously positive score.

**Key Questions For Authors:**

- Can the tail decay condition be formalized as an assumption? Specifically, does AS2 (Lipschitz continuity of $X$) combined with the Schwartz-space condition on $W$ in AS1 imply $X \in L^2)$, or is this genuinely an independent requirement? If independent, it should appear as AS4.

- How does $C(H)$ scale with $L$ and $R$ explicitly? For typical GCN depths (e.g., $L = 3$, $R = 2$), does the bound remain quantitatively informative, or does the depth-induced constant dominate the sparsity-driven decay terms $t_m^{-k'}$ and $n^{-1/2}$?

**Limitations:**

No. See weaknesses.

**Strengths And Weaknesses:**

Strengths:

- The generalized graphon framework is a natural and well-motivated tool for handling arbitrary sparsity, and the paper's use of the double-convergence structure to decouple density and size errors is conceptually clean.
- The construction of sparse graphon sequences via truncation and rescaling is intuitive, and the convergence under the stretched form is illustrated well.
- The experimental evaluation is solid, corroborating the theoretical predictions across datasets of varying scale and sparsity, and the multi-architecture evaluation (varying layers and hidden dimensions) adds robustness.
- The related work is very comprehensive.

Weaknesses:

- The vanishing tail condition is not explicitly flagged as a requirement.

Theorem 5.1 and Lemma 5.2 both contain the term $L^2_X(t_m)$. For the transfer bound to vanish, this tail integral must go to zero, which requires $X \in L^2$ with sufficient decay. This is a non-trivial condition on the signal that is never stated as a formal assumption, never highlighted as a requirement in the main text, and never discussed as a limitation. A reader could easily conclude that the bound vanishes whenever $t_m, n \to \infty$, missing that convergence also depends on the decay of $X$ beyond the truncation horizon. The paper should address this in one of the following ways: (1) add a formal assumption (alongside AS1--AS3) requiring $X \in L^2$ (and explicitly acknowledging this limitation), or equivalently that $L^2_X(t_m) \to 0$ as $t_m \to \infty$; or (2) add a clearly labeled remark following Theorem 5.1 discussing what signal classes satisfy this condition and what happens when it fails.

- The dependence of the bound on filter order and network depth is unaddressed.

The constant $C(H)$ in Theorem 5.1 grows with network depth $L$ and filter order $R$, as is evident from the recursive construction in the proof. It grows at least linearly in $L$. This is the standard cost of spatial-domain analysis and can make the bound loose for deeper architectures. More importantly, the paper does not discuss this in relation to the spectral-domain perspective of Ruiz et al. (2023), where Lipschitz continuity of filters in the spectral domain, specifically near zero eigenvalues, provides tighter control independent of depth. The authors should add a discussion that (1) acknowledges the $L$- and $R$-dependence of $C(H)$ and its practical implications; and (2) positions the spatial approach relative to spectral alternatives, clarifying what is gained (arbitrary sparsity, explicit density decomposition) and what is potentially lost (tightness in depth).

---

> ### Author Rebuttal · Authors · 2026-03-31
>
> We thank reviewer b2Wc for the constructive and thoughtful feedback. Your insights regarding the vanishing tail condition and the influence of architectural parameters improve the rigor and clarity of our theoretical presentation.
>
> ---
>
> > `W1 & Q1`: **The vanishing tail condition is not explicitly flagged as a requirement.**
>
> **R1:** Thank you for the keen insight. $X _ { \mathbb{ R } _ + } \in L^2$ is indeed required to ensure the tail integral $L^2 _ X (t _ m) = \int _ {t _ m}^{+\infty} X^2(u) du$ vanishes as $t_m \to \infty$. And it is independent of AS1 and AS2. Physically, this condition implies that the underlying signal function $X _ {\mathbb{R} _ +}$ has finite energy. Although we briefly mention this requirement when defining the generalized graphon and signal, it was not explicitly presented as a formal assumption. To ensure theoretical rigor, we will incorporate this into the assumption for $X _ {\mathbb{R} _ +}$, clarify the physical meaning of finite energy, and  provide explanation for $L _ X^2(t _ m)$.
>
> ---
>
> > `W2 & Q2`: **The dependence of the bound on filter order and network depth is unaddressed.**
>
> **R2:** Thank you for pointing this out.
>
> For constant coefficients in the error bounds, the spectral analysis by Ruiz et al. (2023) set the maximum filter response to a normalized value of $C_h = 1$, provides a coefficient of $LF^{L-1}$ for depth $L$ and feature channels $F$.
>
> Our analytical approach does not assume filter normalization; the maximum response of filters is $C_h = \max ( \sum_{r=0}^R h_r r, \sum_{r=0}^R h_r ) \sim O(R^2)$. Consequently, our constant coefficient can be represented overall as $L(F R^2)^{L-1}$.
>
> The higher depth $L$ and more hidden channels $F$ lead to larger transfer error, as noted in Ruiz et al. (2023). When filters are not normalized, the higher filter order $R$ also leads larger error values, that is approximately $R^{2(L-1)}$ times compared to the normalized case. Generally, the filter order $R$ is not set very high (e.g., $R=2$) , while hidden channels $F$ is often set higher (e.g., $F=64$), making $L$ and $F$ relatively dominant factors.
>
> Crucially, these coefficients remain **fixed** as the graph size increases and edge density decreases; therefore, the final transfer error converges to zero.
>
> ---
>
> We hope these clarifications address your concerns and we thank you again for your time and expertise.

---

> > ### Author Rebuttal · Reviewer_b2Wc · 2026-04-02
> >
> > Thanks for engaging.
> >
> > Crucially, these coefficients remain fixed as the graph size increases and edge density decreases; therefore, the final transfer error converges to zero. -> Agreed, but isn't the point of this paper non-asymptotic error bounds? Constants matter here.

---

> > > ### Author Response · Authors · 2026-04-03
> > >
> > > Dear Reviewer b2Wc,
> > >
> > > We sincerely thank you for your professional insights and the constructive discussion. We agree with your viewpoint about the constants. *We included the phrase "transfer error converges to zero" in our response primarily for the sake of completeness.*
> > >
> > > Furthermore, we consider the previous discussion, regarding how the constants depend on network depth, the number of hidden channels, and filter order, to be **a highly valuable addition**. We will certainly incorporate these details, alongside other discussed points, into the updated paper.
> > >
> > > Thank you again for your professional and constructive feedback, which has been deeply encouraging to our work.
> > >
> > > Best regards,
> > >
> > > The Authors of Submission 15419

---

### Decision · Program_Chairs · 2026-04-30

**Decision:**

Accept (regular)

**Comment:**

This paper addresses the size transferability of Graph Convolutional Networks (GCNs) across varying sparsity levels by introducing the Generalized Graphon Convolutional Network (GWCN) framework. Utilizing a stretching mechanism, the authors construct a non-trivial limit for sparse graphs and derive explicit transfer error bounds dependent on graph size and edge density.

**Strengths:**

- The generalized graphon framework successfully overcomes the limitations of classical graphon analyses, extending theoretical guarantees to the arbitrary sparsity regimes found in real-world networks.
- The derived transfer error bounds are explicit and interpretable, cleanly decoupling the impact of graph size from expected edge density.
- The empirical evaluation robustly validates the theoretical claims on real-world datasets like Cora, PubMed, and Ogbn-Arxiv across multiple extreme-sparsity trajectories.

**Weaknesses:**

- The theoretical bounds rely on strong mathematical requirements, such as the generalized graphon belonging to the Schwartz space and signals requiring a finite-energy vanishing tail condition, which may restrict practical applicability to heavy-tailed networks.
- The error bound's dependence on network depth and filter order introduces constants that can grow significantly for deeper architectures, which is not thoroughly contrasted with spectral-domain alternatives.
- The experimental setup utilizes a deterministic degree-ordered sampling heuristic that may induce structural biases compared to standard stochastic downsampling methods like random walks.

**Justification:** The submission provides a significant theoretical advancement for GCN transferability by rigorously formalizing transfer error bounds across arbitrary sparsity levels. While reviewers correctly noted limitations regarding stringent mathematical assumptions and the abstraction of the theoretical stretching operation from practical normalization techniques, the core contribution remains highly valuable. The authors provided adequate clarifications during the rebuttal regarding the $L^{2}$ integrability conditions and the impact of architecture depth on the derived constants. Because the framework successfully bridges rigorous graph theory with the practical bottleneck of deploying models on massive sparse networks, the foundational merits of the work outweigh its current methodological limitations.